



# Measurement Report: Long-range transport and fate of DMS-oxidation products in the free troposphere derived from observations at the high-altitude research station Chacaltaya (5240 m a.s.l.) in the Bolivian Andes

Wiebke Scholz[1,*], Jiali Shen[2,*], Diego Aliaga[2], Cheng Wu[3,10], Samara Carbone[5], Isabel Moreno[4], Qiaozhi Zha[2], Wei Huang[2], Liine Heikkinen[2,3], Jean Luc Jaffrezo[6], Gaelle Uzu[6], Eva Partoll[1], Markus Leiminger[1], Fernando Velarde[4], Paolo Laj[6,2], Patrick Ginot[6], Paolo Artaxo[7], Alfred Wiedensohler[8], Markku Kulmala[2], Claudia Mohr[3], Marcos Andrade[4,9], Victoria Sinclair[2], Federico Bianchi[2], and Armin Hansel[1]

[1]Institute for Ion and Applied Physics, University of Innsbruck, Innsbruck, Austria
[2]Institute for Atmospheric and Earth System Research /Physics, University of Helsinki, Helsinki, Finland
[3]Department of Environmental Science and Bolin Centre for Climate Research, Stockholm University, Stockholm, Sweden
[4]Laboratory for Atmospheric Physics, Institute for Physics Research, Universidad Mayor de San Andrés, La Paz, Bolivia
[5]Federal University of Uberlândia, Uberlândia, MG, Brazil
[6]University Grenoble Alpes, CNRS, IRD, INP-G, IGE (UMR 5001), 38000 Grenoble, France
[7]Institute of Physics, University of Sao Paulo, Sao Paulo, SP, Brazil
[8]Experimental Aerosol and Cloud Microphysics, Leibniz Institute for Tropospheric Research, Leipzig, Germany
[9]Department of Atmospheric and Oceanic Sciences, University of Maryland, College Park, MD 20742
[10]Department of Chemistry and Molecular Biology, Atmospheric Science, University of Gothenburg, SE-412 96 Gothenburg, Sweden
[*]These authors contributed equally to this work.

**Correspondence:** Wiebke Scholz (Wiebke.Scholz@uibk.ac.at) and Armin Hansel (Armin.Hansel@uibk.ac.at)

**Abstract.**

Dimethyl sulfide (DMS) is the primary natural contributor to the atmospheric sulfur burden. Observations concerning the fate of DMS oxidation products after long-range transport in the remote free troposphere are, however, sparse. Here we present quantitative chemical ionization mass spectrometric measurements of DMS and its oxidation products H2SO4, MSA, DMSO, DMSO2, MSIA, MTF, CH3S(O)2OOH and CH3SOH in the gas-phase as well as measurements of the sulfate and methanesulfonate aerosol mass fractions at the Global Atmosphere Watch (GAW) station Chacaltaya in the Bolivian Andes located at 5240 m above sea level (a.s.l.).

DMS and DMS oxidation products are brought to the Andean high-altitude station by Pacific air masses during the dry season after convective lifting over the remote Pacific ocean to 6000-8000 m a.s.l. and subsequent long-range transport in the free troposphere (FT). Most of the DMS reaching the station is already converted to the rather unreactive sulfur reservoirs dimethyl sulfone ($DMSO_2$) in the gas phase and methanesulfonate ($MS^-$) in the particle phase, which carried nearly equal amounts of sulfur to the station. The particulate sulfate at Chacaltaya is however dominated by regional volcanic emissions during the time of the measurement and not significantly affected by the marine air masses. In one of the FT events, even some DMS was



observed next to reactive intermediates such as methyl thioformate, dimethyl sulfoxide, and methane sulfinic acid. Also for this

event, backtrajectory calculations show, that the air masses came from above the ocean (distance >330 km) with no local surface contacts. This study demonstrates the potential impact of marine DMS emissions on the availability of sulfur-containing vapors in the remote free troposphere far away from the ocean.

## 1 Introduction

**Dimethyl sulfide, the climate gas.** Dimethyl sulfide (DMS), formed foremost from dimethylsulfoniopropionate (DMSP), that is produced by phytoplankton, is the primary natural contributor to atmospheric sulfur (Bates et al., 1992a; Simó, 2001; Lana et al., 2011). The estimated global DMS flux ranges from about 18 to 34 Tg S year$^{-1}$(Lana et al., 2011), which accounts for half of the natural global atmospheric sulfur burden (Simó, 2001). Undergoing a series of oxidation steps initiated by hydroxyl (OH), halogen, and nitrate (NO$_3$) radicals, DMS contributes to aerosol particle formation and growth and thus to the formation

of cloud condensation nuclei via its oxidation products sulfuric acid (H$_2$SO$_4$) and methane sulfonic acid (CH$_3$S(O)(O)OH, MSA) (Bates et al., 1992a; Charlson et al., 1987; Ayers et al., 1996). It has been estimated that DMS emissions contribute to 18-42% of the global atmospheric sulfate aerosol. Therefore, more direct observations of DMS and also its oxidation products are needed to better understand the effects of DMS on climate (Szopa et al., 2021).

**Oxidation of DMS.** The DMS oxidation takes place via H-abstraction or OH-addition. The H-abstraction and subsequent fast O$_2$-addition leads to the peroxyradical CH$_3$SCH$_2$O$_2^\cdot$. When CH$_3$SCH$_2$O$_2^\cdot$ reacts with RO$_2$ or NO, it forms - via intermediates - the previously mentioned H$_2$SO$_4$ and MSA, participating in new particle formation and growth and CCN formation processes (Covert et al., 1992; Beck et al., 2021). Both products have been detected many times in the marine atmosphere with a wide range of concentrations (Mauldin III et al., 1999; Berresheim et al., 2002; Bardouki et al., 2003; Baccarini et al., 2020;

Beck et al., 2021). The reaction of CH$_3$SCH$_2$O$_2^\cdot$ with HO$_2$ forms a hydroperoxide with the sum formula CH$_3$SCH$_2$OOH (Barnes et al., 2006). Recently, the autoxidation of CH$_3$SCH$_2$O$_2^\cdot$ was theoretically proposed and proven in lab experiments (Wu et al., 2015; Berndt et al., 2019; Ye et al., 2021). This pathway leads to the formation of hydroperoxy methyl thioformate (HPMTF; HOOCH2SCHO), that exists in concentrations of up to 50 pptv ($1.25 \cdot 10^9$ molecules cm$^{-3}$) in the marine boundary layer (Veres et al., 2020). Other products of the abstraction channel are methyl thioformate (MTF, CH$_3$SCHO) and methane

sulfonic peroxide (CH$_3$S(O)$_2$OOH), that were detected in lab studies (Ayers et al., 1996; Barnes et al., 2006; Hoffmann et al., 2016; Ye et al., 2021; Berndt et al., 2019) but - to the best of our knowledge - not yet in ambient air.

In the addition channel, the first stable products are dimethyl sulfoxide (CH$_3$S(O)CH$_3$, DMSO) and methane sulfenic acid (MSEA, CH$_3$SOH). It is likely that MSEA reacts quickly with O$_3$ to form SO$_2$ (Berndt et al., 2020). DMSO reacts with OH, forming methane sulfinic acid (CH$_3$S(O)OH, MSIA) alongside dimethyl sulfone (DMSO$_2$). DMSO$_2$ has small chemical loss

rates and has been observed in the marine boundary layer already in 1998 (Berresheim et al., 1998). A recent cruise study over the Arabian Sea (Edtbauer et al., 2020) even reported concentrations of up to 120 pptv ($3.0 \cdot 10^9$ molecules cm$^{-3}$) of DMSO$_2$.



Due to its long chemical lifetime, its formation and wet deposition rate control its concentration.

**DMS and its products in the free troposphere.** In this measurement report, we present data of DMS and its oxidation prod-
ucts in the tropical free troposphere over the Bolivian Andes, therefore we are now focusing only on previous free-tropospheric
measurements. In high altitudes, >5000 m a.s.l., DMS concentrations are typically between 0 and 5 pptv (Simpson et al., 2001;
Thornton et al., 1992), but can drastically exceed 5 pptv close to convectional updrafts (Thornton and Bandy, 1993; Thornton
et al., 1992). Nowak et al. (2001) measured dimethyl sulfoxide ($CH_3S(O)CH_3$, DMSO) concentrations of about 5 pptv in the
marine tropical free troposphere at approximately 1500 m a.s.l.. Mauldin III et al. (1999) and Zhang et al. (2014) reported
very variable gas-phase MSA concentrations, reaching unexpectedly high values up to $1 \cdot 10^8$ cm$^{-3}$ above 8000 m a.s.l. and
$2 \cdot 10^7$ cm$^{-3}$ at 2500 m a.s.l., respectively, under very dry conditions. Both were suggesting MSA evaporation from aerosols as
a likely source. Unfortunately, no pressure or pressure corrections were mentioned in both pubications, so we can not directly
compare our data to these former measurements. Veres et al. (2020) detected rather low concentrations of HPMTF in the upper
troposphere. Other DMS oxidation products such as $DMSO_2$, MSIA or MTF, have - to our knowledge - not been observed
in the free troposphere. This is probably due to their low concentrations caused by dilution, oxidation, and partitioning onto
aerosols and into cloud water, which challenges analytical instrumentation.

**DMS and free tropospheric aerosol particles.** In the tropical upper troposphere over the oceans, new particle formation
(NPF) has been observed by in-situ aircraft measurements (Williamson et al., 2019) and global models suggest that the parti-
cle formation at these altitudes typically involves $H_2SO_4$ (Dunne et al., 2016; Gordon et al., 2017). DMS oxidation is partly
responsible for the involved $H_2SO_4$, especially over remote marine areas and in the vicinity of convective updrafts (Thornton
et al., 1992). Froyd et al. (2009) detected acidic aerosols in the free troposphere at an altitude of 4-12 km that originated from
convective updrafts over the Pacific and remained in the atmosphere for days up to even weeks during the CR-AVE flight cam-
paign south-west of Central America. The aerosols with diameters $\geq 0.5$ $\mu$m, analyzed by laser mass spectrometry (PALMS,
Thomson et al. (2000)), contained a large fraction of methanesulfonate ($MS^-$), which links them to DMS oxidation. In contrast
to MSA, $H_2SO_4$ can also have other sources, such as anthropogenic or volcanic $SO_2$, which reacts with OH in the presence of
water vapor to form $H_2SO_4$.

**The measurement location.** Our measurements took place at the high-altitude global atmosphere watch (GAW) station at
mount Chacaltaya (CHC), located in the Bolivian Andes at 5240 m a.s.l.. Particle formation and growth events occur regu-
larly at CHC during daytime (onset ca. 10 am local time) with the highest NPF frequencies in May at the beginning of the
dry season (Rose et al., 2015). Especially under westerly wind conditions nearly 100% of the days had particle formation
events. Rose et al. hypothesized that particle formation occurs after the mixing of clean oceanic free tropospheric air masses
with continental air from up-slope winds and convective updrafts. They suggest that the former provides a low-condensation-
sink background thus favoring particle formation and the latter contains the required condensable vapors. However, gas- and
particle-phase composition data of air masses from the free troposphere and continental boundary layer during the different





seasons and wind directions were missing. Possible sources for such condensible vapors in the south-west to the north-west of the station are volcanic outgassing on the Altiplano (a large semi-arid plateau in the Andes), the close-by metropolitan area La Paz / El Alto, or the 75 km distant southern shores of Lake Titicaca during the day, while the Pacific coast is 330 km away from
the station. The intensive measurement campaign *Southern hemisphere high altitude experiment on particle nucleation and growth* (SALTENA) (Bianchi et al., 2021), lasting from the end of December 2017 to the beginning of June 2018 was planned to improve our understanding of the chemistry behind the observed new particle formation. With the Amazon basin to the east and the open Pacific to the west, the air mass composition reaching CHC in the wet and dry seasons are likely very different, as the prevalent wind direction shifts from the lowlands in the east during the wet-season to the west during dry-season.


**This study.** For this study, we focused our analysis on data obtained in May 2018, when the wind came foremost from the direction of the Pacific Ocean and the Altiplano (south-west to north-west), and blue-sky conditions were prevalent as typical for the dry season at the Bolivian Altiplano. Our aim was to quantify condensible vapors and possibly their precursors in air masses reaching CHC via long-range transport in the tropical free troposphere that originated from the Pacific, using an array of
state-of-the-art instruments. We present simultaneous measurements of DMS and its oxidation products $H_2SO_4$, MSA, DMSO, $DMSO_2$, MSIA, MTF, $CH_3S(O)_2OOH$ and $CH_3SOH$ as well as measurements of sulfate and methanesulfonate aerosol mass fractions at CHC. The variety of sources and air masses influencing air composition at CHC (e.g. Bianchi et al. (2021); Rose et al. (2015); Wiedensohler et al. (2018); Achá et al. (2018)) required a careful distinction of air masses impacted by local emissions and long-range transport, which we present in the results, using strict thresholds for typical markers of local urban
emissions in agreement with the FLEXible PARTicle dispersion model (FLEXPART). We test this distinction by analysing the observations of different instruments in the different air masses. In this context, a measurement of the ion composition of the different air masses in a completely different month but under comparable conditions deserves special mention. We then discuss DMS chemical conversion to its oxidation products during transport, considering both their gas phase and particle phase concentrations. With our measurements at CHC we hope to shed some light on the DMS oxidation and transport in the
free troposphere by contributing with measurements of most DMS oxidation products at such altitude.

## 2  Materials and Methods

### 2.1  Measurement location and potential sources of DMS

The measurements took place at the *Chacaltaya Global Atmosphere Watch (GAW) Station* (CHC) which is located east of the
Altiplano close to the main ridge of the Bolivian Andes (16°21'S, 68°07'W, 5240 m a.s.l., see Fig. B1 in Appendix B) and 330 km away from the Pacific coast. The area is semi-arid and partly covered by snow, especially after precipitation events. The vegetation on the Altiplano is dominated by tufts of hard grass on which local farmers let their llamas graze. The metropolis La Paz / El Alto, located 17 km south of Chacaltaya GAW Station with an elevation ranging from 3300-4100 m a.s.l., is an important source of anthropogenic emissions impacting CHC.





The southern part of Lake Titicaca (Lago Menor) lies about 75 km away to the west/northwest of the station. Lake Titicaca, which is the largest freshwater lake in South America with a surface area of 8372 km$^2$, experienced its first algal bloom in 2015. The algal bloom occurred due to anthropogenically increased nutrient levels after climatologically extreme rain events (Achá et al., 2018). These rain events and subsequent algal blooms will likely become more frequent in the future and impact especially the shallow and nitrogen-rich Lago Menor (Duquesne et al., 2021) at the south end of Lake Titicaca. The algal bloom

in the year 2015 involved green algae of the Carteria species (Achá et al., 2018), known to produce DMSP (Franklin et al., 2010) and DMS (Andreae, 1980). Other lagoons on the Altiplano that might host algae and thus emit DMS lie approximately 1000 km to the south. To evaluate the possible impact of these local potential sources on signals of marine trace gases, we used backward Lagrangian dispersion calculations(Aliaga et al., 2021), which are described in more detail in section 2.4. A description of the continuous long-term measurements from the GAW station that we used in this study can be found in section

125 2.3.

## 2.2    The SALTENA campaign

Advanced field measurements with an extensive suite of novel gas- and particle composition instruments at CHC were conducted between December 2017 and the beginning of June 2018 (SALTENA, Bianchi et al. (2021)). The aim of the campaign

was to determine the chemical species of anthropogenic and biogenic origin impacting the station and contributing to new particle formation.

Atmospheric ions were detected by an Atmospheric Pressure interface-Time of flight mass spectrometer (positive-APi-TOF, section 2.2.6), and the gas-phase composition by a nitrate chemical ionization atmospheric pressure interface time-of-flight mass spectrometer (nitrate-CIMS, section 2.2.1), a filter inlet for gases and aerosols (FIGAERO) coupled to a high-resolution

time-of-flight chemical ionization mass spectrometer using iodide chemical ionization (I$^-$-FIGAERO-CIMS, section 2.2.4), as well as a proton transfer reaction time-of-flight mass spectrometer (PTR3, section 2.2.2). An Aerosol Chemical Speciation Monitor (Q-ACSM, section 2.2.5) and the I$^-$-FIGAERO-CIMS further analyzed the aerosol particle composition. Table 1 summarizes which of the analyzed species are detected by which intrument and an overview of the operation times of the instruments and the analyzed period is shown in Fig. 1.

**Table 1.** An overview over the mass spectrometers during the SALTENA campaign and their respective detected species

| Instrument | detected species | gas-phase | particle-phase |
|---|---|---|---|
| Q-ACSM | summed organics, nitrate, sulfate, ammonium, chloride | | x |
| I$^-$-FIGAERO-CIMS | MSA, speciated oxidized organics | x | x |
| PTR3 | DMS, DMSO, DMSO$_2$, MSIA, CH$_3$SCHO, CH3SOH, speciated organics | x | |
| nitrate-CIMS | MSA, H$_2$SO$_4$, CH$_3$S(O)$_2$OOH, speciated highly oxidized organics | x | |
| Positive-APi-TOF | ionic clusters of DMSO, amines, water, organics | x | |



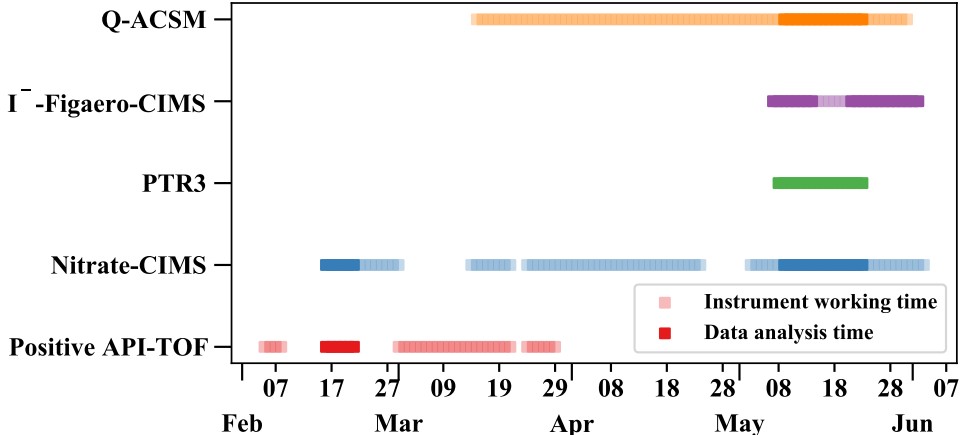

**Figure 1.** Overview over the operation times of the individual mass spectrometers and the analyzed time periods

### 2.2.1 nitrate-CIMS

The nitrate-CIMS (Tofwerk AG, Thun, Switzerland) measured the concentration of $H_2SO_4$, MSA and a compound with the exact mass of $CH_3S(O)_2OOH$ from April 19th to 25th, and May 5th to June 3rd. The specially designed inlet for chemical ionization at ambient pressure is described by Kürten et al. (2011) and Jokinen et al. (2012). The data we report here are averaged for 10 minutes. The nitrate-CIMS uses nitrate anions $[(HNO_3)_n\ (NO_3^-)$, n = 0-2] as reagent ions to ionize gas molecules (M) via proton transfer from the molecule to the nitrate ions or by or ligand switching reactions, forming $MNO_3^-$ clusters.

The signal of the detected compound in counts per second (cps) is normalized to the sum of count rates of reagent ions and then multiplied with the calibration coefficient $C$. The calibration coefficient $C = 1.5 \cdot 10^{10}$ cm$^{-3}$ was determined after the campaign, using $H_2SO_4$ as calibrant, following the procedure by Kürten et al. (2012). We assume the effect of temperature and humidity on charging efficiency to be negligible. Due to the lack of standards, we could not calibrate every individual species we detected. Therefore, we use the $H_2SO_4$ calibration factor $C$ to estimate MSA and $CH_3S(O)_2OOH$ concentrations. The sampling flow was 10 slpm, mixed with 20 slpm sheath flow before measurement. The inlet is a 150 cm long stainless steel tube with a 3/4 inch inner diameter, and its line loss for $H_2SO_4$ is already included in the calibration factor.

### 2.2.2 PTR3

The PTR3 ionizes sample gas molecules with $(H_2O)_nH_3O^+$ clusters mainly via ligand switching ($H_3O^+$-mode), and detects volatile organic compounds (VOCs) and their oxidation products. It has been described previously by Breitenlechner et al. (2017). During the SALTENA campaign, the length and diameter of the inlet were 90 cm and 10 mm, respectively. Core sampling right before the ion-molecule reaction region, subsampling only 1.2 out of 7.5 slpm air, reduces inlet wall losses to approximately 55% as calculated from laminar transport and radial diffusion. From the 8th of May to the 23rd of May 2018





we used an RF amplitude of 400 Vpp at a pressure of 55 mbar and a temperature of 310 K, corresponding to an E/N (E: electric field and N: concentration of neutral particles) of $106 \pm 25$ Td, leading to primary ion distributions as shown in Fig. C1 in Appendix C1. Compounds with proton affinities above $\sim 200$ kcal/mol, like hexanone or DMSO, are ionized at the kinetic limit. Those with lower proton affinities need to be calibrated individually, which is e.g. the case for DMS. During the campaign we calibrated the PTR3 regularly by mixing 7.5 sccm of nitrogen 5.0, containing 1 ppm of hexanone (98%), 1,2,4-

trimethyl benzene (98%), alpha-pinene ($\geq 99.9\%$) and acetonitrile ($\geq 99.9\%$), each, as reference standards (all purchased from Sigma-Aldrich), into 7.5 slpm of compressed air with purity 5.0. Hexanone is our reference for compounds ionized at the limit of detection in the $H_3O^+$-mode and gives us the maximally possible sensitivity for a sample gas molecule. To lower the limit of detection, we averaged the data to a 5-minute time resolution.

### 2.2.3 Quantification of gas-phase DMS and its oxidation products detected by PTR3 and nitrate-CIMS

For DMS, detected by the PTR3, we performed water-vapor-mixing-ratio-dependent off-site calibrations after the campaign together with acetonitrile (nearly daily on-site calibrations available) as a reference. The calibration curves are shown in Fig. C2 in Appendix C1. The DMS time series was then calibrated, using the humidity-dependent calibration function, multiplied by a time-dependent transmission factor that we inferred from the comparison of the regular acetonitrile calibrations at their respective humidity with its humidity-dependent calibration curve. For DMS (calibrated), MSIA, DMSO, and $DMSO_2$ (ionized

at the kinetic limit in $H_3O^+$-mode, see Shen et al. (2022)) we have good references and can therefore assume a low uncertainty of the concentrations of around 30%. For all other oxidized organosulfur compounds observed by the PTR3 of which proton affinities are typically unknown, we assumed a sensitivity at the kinetic limit, so that their presented concentrations are lower-limit estimates.

Analogous to $H_2SO_4$, MSA, detected by the nitrate-CIMS, has a collision-limited charging efficiency when reacting with the
nitrate ions resulting in strongly bound clusters not prone to fragmentation. It is therefore valid to use the $H_2SO_4$ calibration factor. However, we can only give the lowest estimated concentration for $CH_3S(O)_2OOH$ due to its low detection efficiency (Shen et al., 2022).

Small signals close to the detection limit or background (e.g. due to neighbouring peaks) make the quantification more difficult. To discuss the impact of the limit of detection on the data quality of some important compounds, we present lower limit

concentration distributions of detected organosulfur compounds from PTR3 and nitrate-CIMS together with their limits of detection in Fig. C3 in Appendix C2. The limit of detection takes also the maximum amount of interference by neighbouring peaks into account. Apart from DMSO and $DMSO_2$, all of the compounds frequently exhibited low concentrations close to the detection limit so their concentrations are somewhat uncertain during such times. In our analysis, however, we focus on periods when the signals were typically higher than their median and thus higher than their respective limit of detection. The

peaks are clearly visible and quantifiable under these circumstances. All concentrations reported are given in molecules cm$^{-3}$ for standard conditions.



### 2.2.4 I⁻-FIGAERO-CIMS

The I⁻-FIGAERO-CIMS (Aerodyne Research) was deployed to measure the molecular composition of both gas-phase and particle-phase organic compounds and inorganic acids. The FIGAERO inlet has two modes. In the gas-phase mode, ambient air was directly sampled into the ion-molecule reactor while particles were simultaneously collected on a 25mm Zefluor poly-tetrafluoroethylene filter through another sampling port with a flow of 3.8 slpm. The duration of particle-phase sampling was 120 minutes. When the gas-phase measurement (particle-phase sampling) was completed, the FIGAERO inlet was switched to the particle-phase mode and a nitrogen gas stream (2 slpm) was heated and blown through the filter to evaporate the particles via temperature-programmed desorption. More details about the instrument can be found in Lopez-Hilfiker et al. (2014) and Thornton et al. (2020). The instrument was working from the 10$^{th}$ of April to the 2$^{nd}$ of June 2018 with only short interruptions due to power outages. From the 14$^{th}$ to the 21$^{st}$ of May no particle-phase, but only gas-phase data are available.

### 2.2.5 Q-ACSM

A Quadrupole Aerosol Chemical Speciation Monitor (Q-ACSM, Aerodyne Research Inc.) (Ng et al., 2011) measured the non-refractory submicron aerosol components: organics, nitrate, sulfate, ammonium, and chloride. To remove larger particles to avoid clogging the Q-ACSM inlet, a $PM_{2.5}$ cyclone was integrated in the inlet. The Q-ACSM was calibrated with ammonium nitrate and sulfate, following the recommendations of Ng et al. (2011), who also described the instrument in further detail. The time resolution of the Q-ACSM was 30 minutes. In this study, we use the data from the Q-ACSM collected during 9–23 May 2018.

### 2.2.6 Positive-APi-TOF

The Atmospheric Pressure interface-Time of Flight (APi-TOF) mass spectrometer was operated in positive ion mode to detect positively charged ions and ion clusters from February to March 2018. From April to June, we switched the Positive-APi-TOF to the negative mode for measuring negatively charged ions and ion clusters. A detailed description of the instrument and operation can be found in Junninen et al. (2010). We used 20-minute averages when pre-processing the data. The total inlet flow was kept constant at $\sim 14.5$ slpm of which 0.8 slpm were analyzed by the instrument to minimize losses of ions to the inlet walls. The inlet line was a stainless steel tube of 1 cm inner diameter, and $\sim$100 cm length.

### 2.3 Long term measurements at CHC

The data from the SALTENA campaign were analyzed in the context of the data from the continuous measurements taking place at the GAW station, which is equipped with standard instruments to measure trace gases like CO and $CO_2$ for example. An Automatic Weather Station (AWS) deployed at the station records air temperature, relative humidity, radiation, wind direction, and wind speed at a 1-minute resolution. Aerosol physical, chemical, and optical properties have been measured nearly continuously in the past ten years. The aerosol particle size distribution is monitored with a Mobility Particle Sizer Spectrometer (MPSS), assembled with a TSI Inc. 3772 Condensation Particle Counter and a custom-made Differential Mobility Analyzer.





The MPSS measures the particle size distribution in the range of 10 nm to 650 nm with a five-minute time resolution. A Multi Angle Absorption Photometer (MAAP) monitors the equivalent black carbon (eBC) mass with a time resolution of one minute

(Petzold et al., 2005). The data are reported at standard temperature and pressure conditions, following the GAW network recommendations (WMO and GAW, 2016), and the data are corrected considering the effective wavelength of the instrument in accordance with Müller et al. (2011).

Two interchangeable digital impactors are used to collect the aerosol particulate matter with aerodynamic diameters lower than 10 and 2.5 $\mu$m (i.e., $PM_{10}$ and $PM_{2.5}$, respectively). The collectors are equipped with pre-baked quartz fiber filters (Ø 150 mm)

collecting samples at a flow rate of 30 $m^3$ $h^{-1}$. The filters are exchanged once a week and are analyzed by a Dionex Ion Chromatograph, Sunset Lab TOC (applying the EUSSAAR2 protocol), and High-Performance Liquid Chromatography coupled to a Pulse Amperometric Detector to determine the ions, organic and elemental carbon, and anhydrous monosaccharides. We use data of methanesulfonate from December 2011 to August 2017. A more detailed description of the site and further long-term measurements can be found in previous studies (Wiedensohler et al., 2018).

**2.4 Analysis of air mass history**

Aliaga et al. (2021) used a high-resolution meteorological model coupled with Langrangian dispersion calculations to identify where air masses sampled at CHC originate from, and to quantify the relative influence of the surface and the free troposphere. Here we utilise, and further analyze, the output from the same simulations.

Aliaga et al. (2021) performed a 6-month-long simulation with the Weather Research and Forecast (WRF) model. The simu-

lation had 4 nested domains. The outermost and largest domain had a grid spacing of 38 km, whereas the innermost domain, covering the area closest to CHC had a grid spacing of 1km. The meteorological output of the WRF simulation was saved every 15 minutes and used as input to the Lagrangian dispersion model FLEXPART (FLEXible PARTicle dispersion model). The high temporal and spatial resolution of the driving meteorological data is a key advantage over dispersion or trajectory simulations driven by reanalysis data - typically only available every 1 to 6 hours. Backward dispersion simulations were performed

with FLEXPART. For every hour, 20,000 particles were randomly distributed and then released from a 10 m deep (a.g.l) layer covering a 2 x 2 km square centered on CHC. The trajectories of all particles were tracked back in time for four days. It should be noted that the particles were treated as passive tracers that do not undergo any chemical or microphysical processing.

The output of FLEXPART is the source-receptor-relationship (SRR), which is related to the particles' residence time in the 3-dimensional grid cells of the FLEXPART output grid. Aliaga et al. (2021) transformed the SRR output from the Cartesian grid

to a log-polar grid so that the size of the grid cells gradually increases with the distance from CHC (the receptor); additional details of this transformation, and its motivation, are provided in Aliaga et al. (2021). The domain of the log-polar grid is a cylinder covering an area with a radius of 1600 km centered on Chacaltaya and extending from the surface to 15 km a.g.l..

As the FLEXPART domain is limited in size, it is possible that during the four days of backwards travel particles can leave the domain. Therefore, unlike in Aliaga et al. (2021), we also estimated the amount of air masses arriving at CHC originating

from outside the FLEXPART domain (herein referred to as "out of domain"). Similar to Aliaga et al. (2021), we identified air masses arriving from different pseudo layers within the FLEXPART domain (referred to as "within domain"): Surface layer





(air masses from the lowest FLEXPART output layer 0-500 m a.g.l.), boundary layer (air masses from below 1500 m a.g.l.), and free-troposphere (air masses from above 1500 m a.g.l.).

The out-of-domain air mass contribution is computed as follows: Theoretically, if all particles remained in the domain, the domain-integrated SRR would be 4 days. However, in the case when particles leave the domain, the domain-integrated SRR is less than 4 days. The out-of-domain SRR is therefore computed as the theoretical maximum SRR minus the actual SRR. No information is available about the trajectories of the particles once they have left the domain.

**Footprint analysis.** To identify the specific source regions of chemical species measured at CHC, the SRR from FLEXPART can be combined with the in-situ measurements of such species via an elastic net regression (Pedregosa et al. 2011). The same method was used to identify likely sources of black carbon measured at CHC (Bianchi et al., 2021). In contrast to Bianchi et al. (2021), however, we included also the lateral boundaries of the FLEXPART domain in the footprint analysis to visualize the long-range transport into the domain through these boundaries.



## 3 Results

### 3.1 Determining free tropospheric periods under typical dry season conditions

To identify periods dominated by free tropospheric air masses versus periods dominated by boundary layer emissions, we present a compilation of important measurements (all reported for standard conditions) in Fig. 2. The time series of gas-phase MSA is shown in Fig. 2 A), as well as the sum of $C_8H_{10}O_x$ compounds, which are likely oxidation products of aromatics xylenes or ethylbenzene, and we can thus use them as indicators for anthropogenic emissions (e.g. Li et al. (2019)). It is noticeable, that gas-phase MSA and $C_8H_{10}O_x$ compounds appear anticorrelated.

We also present Q-ACSM total particle mass and its composition (Fig. 2 E), the particle number size distribution (Fig. 2 F), water vapor mixing ratio (WVMR) and equivalent black carbon mass (eBC) (Fig. 2 C), as well as short wave radiation (SWR), and temperature (Fig. 2 D). Fig. 2 B shows the amount of air coming from the free troposphere ($> 1500$ m a.g.l.), and influenced by surface emissions (0-500 m a.g.l.) according to the results of the FLEXPART lagrangian dispersion analysis (see Aliaga et al. (2021), section 4 for a justification of the differentiation using constant pseudo layer heights instead of a time-resolved boundary layer height). Neglecting the varying boundary layer height in these traces leads to a tendency to underestimate the surface influence during daytime and the free tropospheric air during nighttime (when the inversion is below 1500 m a.g.l.). Furthermore, we show the fractions of free tropospheric air from "out of" the FLEXPART domain and "within the domain" (defined in sec. 2.4).

Fig. 2 B shows that during the period shown, free tropospheric air masses reach the CHC GAW station all the time (see also Aliaga et al. (2021)). Especially during the daytime, free tropospheric air mixes with local air masses from the surface layer (spikes in the surface layer influence in Fig. 2 B). This leads to enhanced concentrations of the anthropogenically emitted C8 compounds (Fig. 2 A) as well as a larger nitrate fraction in the particle mass (Fig. 2 E). We observe particle formation events nearly every day as previously reported by Rose et al. (2015), typically followed by particle growth that regularly extends into the night and sometimes without interruption (e.g. 15th-16th, 16th-17th of May 2018) until the next particle formation event the next day (Fig. 2 F). However, during certain periods (marked by blue shadings in Fig. 2), at late night or early morning, when the temperature reaches its daily minimum, eBC (Fig. 2 C) and the particle mass determined by the Q-ACSM (Fig. 2 E) show a sudden drop, coinciding with overall lower particle numbers measured by the MPSS (Fig. 2 F, see also Fig. D2 in Appendix D1 for the number size distribution), and particularly low water vapor mixing ratios (Fig. 2 C). These are strong indications that the station was above the shallow boundary layer of the Altiplano and that free tropospheric air masses had subsided from higher altitudes.

The FLEXPART dispersion simulation does clearly resolve this: For the short event in the morning of the 11[th] of May, for example, Fig. 3 C and D clearly show, that the air reaching CHC descended from higher altitudes and had not been in contact with the surface before detection (<5% of the air had been in contact with the surface according to FLEXPART). In contrast, during the afternoon of the same day, shown in Fig. 3 B1 and B2, a still small, but larger fraction of the air travelled uphill close to the surface (ca. 10% of the air had been in contact with the surface according to FLEXPART). During the whole day, the main wind direction was west-southwest, as shown in Fig. 3 A and B for the 8 am case. To summarize, a shallow and



**Figure 2. Overview of two weeks of data for the identification of free tropospheric periods.** Methanesulfonic acid (MSA) and summed $C_8H_{10}O_x$ (A), the influence of surface ($< 500$ m a.g.l.) and free tropospheric air masses ($> 1500$ m a.g.l.) according to FLEXPART, with free tropospheric (FT (total)) air masses shown additionally divided into the air from within and without the FLEXPART domain (B), water vapor mixing ratio (WVMR, solid) and equivalent black carbon mass (eBC, dashed) (C), short wave radiation (SWR) and temperature (T) (D), particle matter (PM) mass and composition from Q-ACSM (E) and particle number distribution ($\frac{dN}{d\log_{10}(D_p)}$) (F) versus local time. Lightblue shaded areas mark times that we consider dominated by free tropospheric air masses due to the low particle load and water vapor mixing ratio.





**Figure 3. Air mass origin comparison for a free tropospheric event (Panels A-D) and one with surface contact of the air masses (Panels E and F).** Panels A and B show the top-down view of the most likely source regions (color-coded in red) showing the origin of the air parcels during the free tropospheric event on the 11[th] of May. Panels C and D show the same data but height resolved together with the topography as a vertical cut through the atmosphere in the direction of most likely origin (west-southwest). For comparison, panels E and F show the most likely source of air parcels during a time with surface contact of the arriving air masses at 2 pm on the same day. Panels B, D, and F present a zoom on the local area close to Chacaltaya that mainly determines the influence of local sources like anthropogenic emissions from La Paz / El Alto.





strong inversion close to the ground likely did not allow the mixing of the different air masses during the marked times. This gives us the unique opportunity to characterize the free tropospheric air masses reaching the station from the westerly sector

(as typical for the Altiplano dry season) with our array of state-of-the-art mass spectrometers. We determine when the station is in the free troposphere above the shallow boundary layer by applying the conditions from Table 2. All periods during which

**Table 2.** Identified threshold values to determine periods with free tropospheric (FT) air from higher altitudes reaching CHC undisturbed by mixing with the local boundary layer during the dry season.

| FT identifier | Threshold | Units |
|---|---|---|
| Water vapor mixing ratio | $<1.0$ | $\text{g kg}^{-1}$ |
| Q-ACSM particle mass | $<0.3$ | $\mu\text{g m}^{-3}$ |
| Equivalent black carbon (eBC) mass | $<0.1$ | $\mu\text{g m}^{-3}$ |

the conditions are not fulfilled, are likely influenced by the local boundary layer to at least some extent and we refer to them as $\text{BL}_{\text{day}}$ between 7 am and 7 pm local time and $\text{BL}_{\text{night}}$ otherwise. The data in the following figures represent averages over 32 hours for each case. Each single $\text{BL}_{\text{night}}$ and $\text{BL}_{\text{day}}$ period was chosen such that they were temporally as close as possible

to and of the same duration as an FT period. Additionally, the main wind direction had to be the same as during the respective FT period to ensure comparability.

Due to the limits of the FLEXPART domain, we cannot know exactly where the free tropospheric air masses had the last contact with the boundary layer, but chances are high that the last surface interaction occurred in a marine, remote setting over the south- to the southwestern Pacific Ocean or in the innertropical convergence zone (ITCZ), as convection typically occurs in

these areas, while it is typically unlikely in the south-eastern Pacific due to the cold Humboldt upwelling. The enhanced gas-phase MSA concentrations during the marked undisturbed free tropospheric events, detected by the nitrate-CIMS and shown in Fig. 2 A), are the first indication of this proposition. We also observed higher-than-median concentrations of MSA outside the marked periods, indicating that free tropospheric air masses mix in regularly. With our FT identification rule, we made sure to select events, that are least impacted by surface emissions. That does not exclude the influence of FT air masses during other

periods.

The Q-ACSM identifies the aerosols in the free troposphere as composed of sulfate and organics with only little ammonium and no nitrates (see also Fig. D2 panel F in Appendix D1). The relative absence of nitrate observed in the particles by the Q-ACSM is reasonable, as it likely links to ammoniumnitrate that forms from traffic emissions in the La Paz / El Alto metropolitan area and is thus solely emitted into the local boundary layer. The large sulfate fraction we observed probably has a regional origin

and is not solely attributed to marine DMS oxidation, because CHC is influenced by volcanic emissions in the area (Bianchi et al., 2021), especially under westerly wind direction.



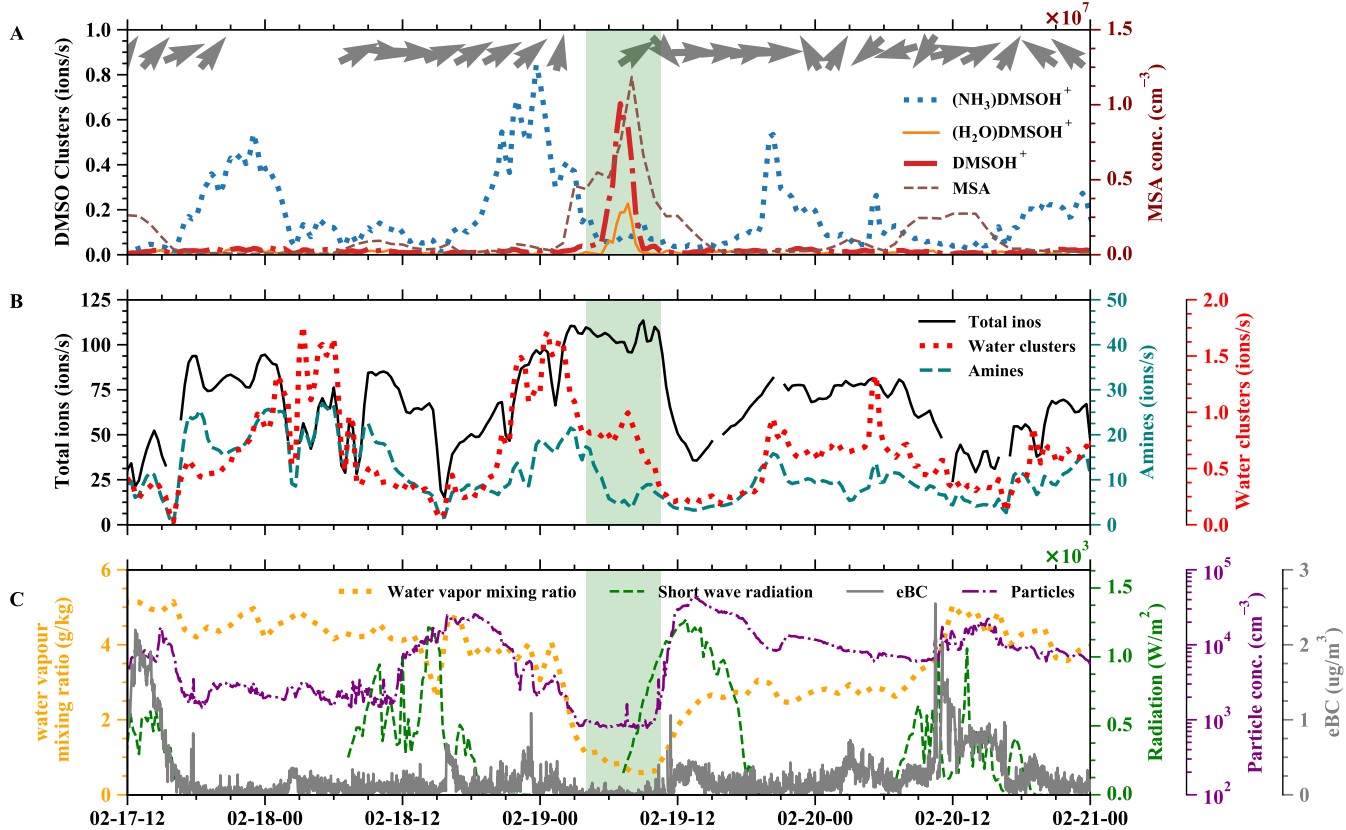

**Figure 4. Time evolution of positive ions before, during, and after an FT event (green shaded).** A) $(NH_3)DMSOH^+$, $DMSOH^+$ and $(H_2O)DMSOH^+$ clusters, with arrows representing the wind direction on top B) Total positive ions (left axis), amines, and water–ammonia clusters (right axis) from positive APi-TOF. C) Water vapor mixing ratio (left axis), radiation, temperature, and equivalent black carbon (eBC) (right axes).

### 3.2 Ionic cluster composition in the free troposphere.

The positive APi-TOF can give us information on the positive ion composition of the free tropospheric air masses. Using our previously gained understanding of the fingerprint of the FT events, we found an FT period under the west-wind influence in February, when the instrument was operating.

In Fig. 4 we show a free tropospheric time period (19[th] of February, 3 am - 10 am) highlighted in green. A detailed air mass origin description for the FT event on 19[th] of February is shown in Fig. E2 in Appendix E, which is close to the conditions of the event on the 11[th] of May.

As described above, the low water vapor mixing ratio (dropping from previously 4 g/kg to 0.8 g/kg), low equivalent black carbon concentration (below $0.1 \mu g\,m^{-3}$), and low total number concentration of particles in the diameter range 10-650 nm (Fig. 4 C) are indications of such FT events. The wind direction measured at the peak of the Chacaltaya mountain is shown as





barbs in Fig.4 A, together with the time series of different DMSO clusters. DMSO has a proton affinity of $884.4\ \mathrm{kJmol^{-1}}$. The signals of the typically dominating amines (proton affinities above $900\ \mathrm{kJmol^{-1}}$) in the positive APi-TOF are significantly smaller during FT than during BL times (see Fig. 4 B), while other compounds (unidentified, likely organics) take most of the 340 charge (see Fig. E1 in Appendix E).

The signals of $\mathrm{DMSOH^+}$ and $\mathrm{(H_2O)DMSOH^+}$ show an increase during the free tropospheric air event. This does not necessarily imply a higher DMSO concentration in the free troposphere; after all, $\mathrm{(NH_3)DMSOH^+}$ was regularly present throughout the whole period (17$^{\mathrm{th}}$-21$^{\mathrm{st}}$ of February) with maxima during nighttime and minima during daytime. This does indicate, however, the general presence of DMSO and additionally the lower abundance of species with higher proton affinities in FT air, 345 which does typically affect the ionization likelihood of species with lower proton affinities (Tanner, 1990). Therefore, the strong signals of $\mathrm{DMSOH^+}$ and $\mathrm{(H_2O)DMSOH^+}$ and the decreased signal of $\mathrm{(NH_3)DMSOH^+}$ show that ammonia, amines, and pyridine are significantly less available during FT times. Even though the water vapor mixing ratio decreased during the FT event, the water concentration in the atmosphere always remains high in contrast to other trace gases, which explains the slight increase of water clusters.

As shown in Fig. 4 B, the signals of DMSO clusters are small compared to the total ions. The total ion counts remain high during the FT event because the total ion concentrations in the atmosphere are determined by solar radiation, altitude, and other parameters. In our case, the ions switch from the species with the highest proton affinity to ones with lower proton affinity during air mass transition, which can explain the increase of the signal of unidentified organics, as shown in E1, and proves a significantly lower availability of high proton affinity compounds such as amines.




### 3.3 Observations of DMS oxidation products in the free troposphere

The molecular formulae of the organosulfur compounds detected in the gas phase by the PTR3 and nitrate-CIMS indicate that they are, in fact, oxidation products of DMS. In particular, we detected all presently known DMS oxidation products, except the recently observed HPMTF (Wu et al., 2015; Berndt et al., 2019; Veres et al., 2020), which might be below the detection

limit.

We show their time series in Fig. D1 in Appendix D and present boxplots of the detected concentrations in the different

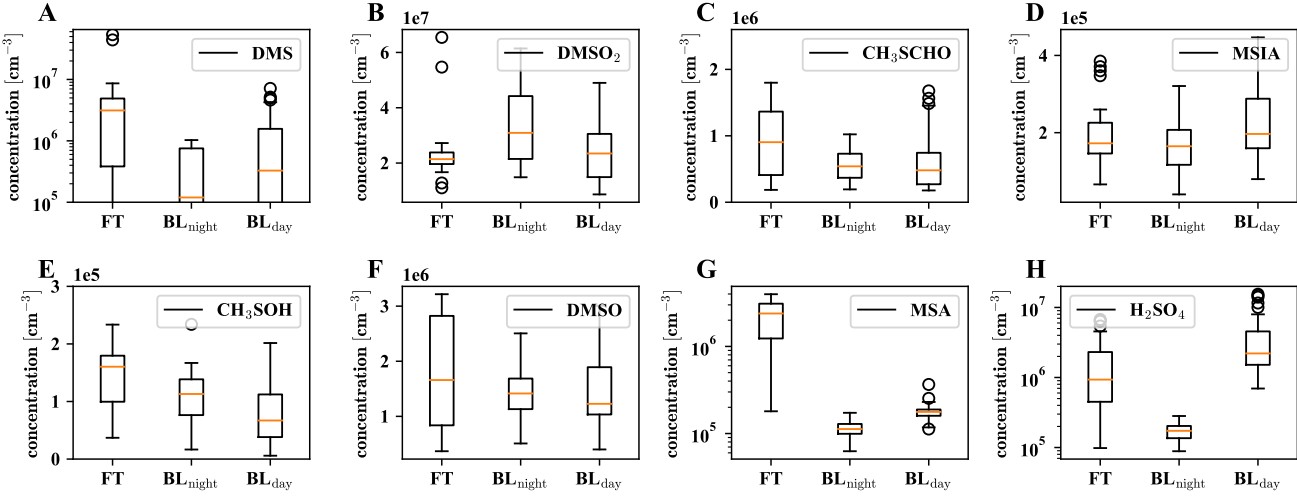

**Figure 5.** Boxplots of gas phase DMS oxidation products, detected by PTR3 and nitrate-CIMS in the FT (free troposphere without boundary layer influence), $BL_{night}$ (within boundary layer during night time), $BL_{day}$ (within boundary layer during day time) air masses, respectively.

air masses in Fig. 5. DMS (Fig. 5 A) and gas-phase MSA (Fig. 5 G) show significantly enhanced concentrations in the free troposphere. MSA is in fact one of the few condensible vapors which are higher in FT air than during $BL_{day}$ and dominates the nitrate-CIMS' FT - $BL_{day}$ spectrum shown in Fig. D3. When comparing with Fig. D1, the sporadic high concentrations

of DMS occur during the first analyzed FT air period on the 11$^{th}$ of May. One rare subtropical storm, which was observed from the 4th to 9th of May only a few 100 km off the Chilean coast (NESDIS, 2018), might have impacted our data and could explain the surprisingly high DMS concentration in such a large distance from the typically convective regions over the southern Pacific substantially further west.

Gas-phase $H_2SO_4$ (Fig. 5 H) is strongly enhanced in the FT compared to the $BL_{night}$, but is maximal in $BL_{day}$. Since $H_2SO_4$

formation is driven by photochemistry, a diurnal maximum in $H_2SO_4$ concentration can be explained, if local production, e.g. from $SO_2$ oxidation occurs, although enhanced losses onto particles during $BL_{day}$ (Fig. D2 panel D in Appendix D1) have a decreasing effect on the $H_2SO_4$ gas-phase concentration. Due to the influence of the metropolitan area of La Paz/El Alto and volcanic emissions, we expect $H_2SO_4$ to have a large OH-initiated production rate by oxidation of anthropogenic and volcanic





SO$_2$ during daytime. The high concentration of H$_2$SO$_4$ in the FT is partly explained by an increase in sulfuric acid directly
after sunrise as some of the FT periods extended into the morning hours. Additionally, the lower condensation sink during the
FT periods enhances the sulfuric acid lifetime during these events. This is especially evident in the night of the 20$^\text{th}$ of May:
Although in the middle of the night, where we expect the production term to be small, a substantial amount of sulfuric acid is
observed that behaves anti-correlated to the low condensation sink during the event, as shown in Fig. D4 in Appendix D2.
The other oxidation products show a significantly weaker trend - but none shows features of a typical boundary layer product
(compare to Fig. D2). However, as seen in Fig. 2, there is always an influence of free tropospheric air. A compound with little
to no trend between those conditions could thus still have its origin in the free troposphere, but in such cases, additional local
sources cannot be excluded and should be checked. We are doing this for MSIA (Fig. 5, panel D) in section 3.5. Afterwards,
we will relate their detected concentrations in section 3.6.





### 3.4 Methanesulfonate and Sulfate

The gas-phase MSA concentration in the FT is an order of magnitude higher than in the BL air (Fig. 5 G), but because of the low vapor pressure of MSA, we expect it to reach CHC mainly in the particle phase. We show the signal of particulate MSA (methanesulfonate), measured by the $I^-$-FIGAERO-CIMS in Fig. 7 A above the time series of the influence of the different air masses according to FLEXPART in Fig. 7 B.

First of all, the FLEXPART analysis shows that the station is under the influence of free tropospheric air throughout the de-
scribed measurement period. The total (mainly organic) signal of the particle phase detected by the $I^-$-FIGAERO-CIMS (grey in Fig. 7) shows strong spikes corresponding to the daily influence of the boundary layer at the station. In contrast to the total signal, the methanesulfonate (light blue) stays relatively constant throughout the week and its fraction in the particles (dark blue, circles) is anticorrelated to the boundary layer influence (25th - 28th of May). From the 29th of May onwards, we observed a higher fraction of the methanesulfonate signal, likely due to the increased fraction of air masses from outside the
domain. Wind velocities are significantly higher in higher altitudes in the free troposphere, so air masses get transported over large distances. On the night of the 30th of May, the fraction of air parcels from the free troposphere reaches nearly 90%. This includes the fraction of air parcels from outside the domain that reached even up to 40% during this period. At this time, also the methanesulfonate fraction in the particles reached its maximum. This strongly suggests that the methanesulfonate reaches CHC via transport in the free troposphere and that there is no significant local source.

As the absolute methanesulfonate signal appears independent of the surface influence, we can also use sampling filter measurements with a very coarse time resolution. While the filter measurements do not allow a differentiation between the free troposphere and boundary layer air masses, they allow for investigating the methanesulfonate more quantitatively. The long-term particle composition measurements using sampling filters are covering the years 2012-2017 (Fig. 6) and show that approximately $4 \pm 1$ ng m$^{-3}$ methanesulfonate is found in the particles from April to November, which corresponds to the time
when west-wind conditions occur regularly. The particulate mass concentration of methanesulfonate corresponds to an average number of roughly $2.5 \cdot 10^7$ molecules cm$^{-3}$, which is 1-2 orders of magnitude higher than the detected gas-phase methane sulfonic acid concentrations, in line with its low vapor pressure.

The strongly FT-dependent gas-phase MSA signal observed from the nitrate CIMS (compared to the nearly constant methanesulfonate signal from the $I^-$-FIGAERO CIMS) could be explained in two ways: Firstly, MSA is known to participate in new
particle formation and growth to larger aerosol particles (Bates et al., 1992b; Ayers et al., 1996; Covert et al., 1992; Beck et al., 2021) and the condensation sink in surface-affected air masses is much higher compared to within free tropospheric air masses. In FT air, the mild condensation sink leads to a longer lifetime for gas-phase MSA, which can explain the higher concentration of gas-phase MSA in the periods of undisturbed free tropospheric air reaching CHC. Upon mixing of the free tropospheric air masses with the boundary layer, the stronger condensational loss of MSA to the aerosols fully scavenges the gas-phase MSA.
Secondly, it was suggested in earlier publications (Mauldin III et al., 1999; Zhang et al., 2014) that MSA might degas from methanesulfonate aerosol in air masses with low relative humidity. The herein analyzed FT events always involved very low relative humidities, so this effect might occur as well. Zhang et al. (2014) stated that parallel measurements of both MSA and



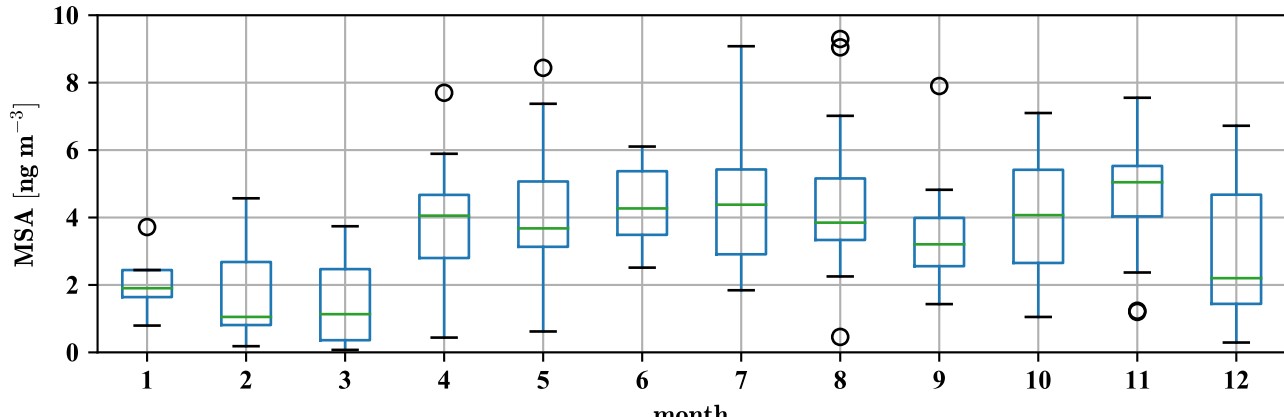

**Figure 6.** Yearly pattern of methanesulfonate mass concentration in PM2.5 and PM10 particles (based on data between 2012 and 2017, taken at CHC). In the months from December to March when the wind is often coming from the east due to the Bolivian high (Bianchi et al., 2021), methanesulfonate mass concentrations in the particles are significantly lower than during the rest of the year.

DMSO are necessary to prove or disprove their MSA evaporation hypothesis. We relate MSA to the other oxidation products in section 3.6 after considering the impact of local sources.

Local sources certainly impact $H_2SO_4$, as already discussed in section 3.3. This is confirmed by comparing methanesulfonate and sulfate concentrations: Sulfate fractions of 50% were detected by the Q-ACSM (Fig. D2 in Appendix D1) even during the marked free troposphere periods, which means that approximately $100\,\mathrm{ng\,m^{-3}}$ of non-refractory PM1 would be made up of sulfate (see Fig. D2, panels E and F).

In comparison, methanesulfonate from the filter measurements is only about $4\,\mathrm{ng\,m^{-3}}$ in May in PM2.5 and PM10, (Fig. 6).

The ratio between methanesulfonate and sulfate is thus below 0.04, which is significantly lower than in remote marine areas (Ayers et al., 1996), laboratory experiments (Shen et al., 2022) or at similarly cold temperatures in free tropospheric measurements of marine air (Froyd et al., 2009). This suggests that the remote marine particle composition is altered due to the volcanic $SO_2$ emissions in higher altitudes prior to reaching CHC as mentioned in Bianchi et al. (2021). While we might be able to limit the impact of the anthropogenic emissions on our dataset by using the FT indicators discussed in section 3.1, we are hesitant to

exclude the volcanic impact on sulfuric acid and sulfate data even in the undisturbed FT periods (defined by the specifications in Table 2) due to the higher emission altitude of volcanoes. Therefore, our dataset does not allow any conclusions for the production of $H_2SO_4$ by DMS oxidation.




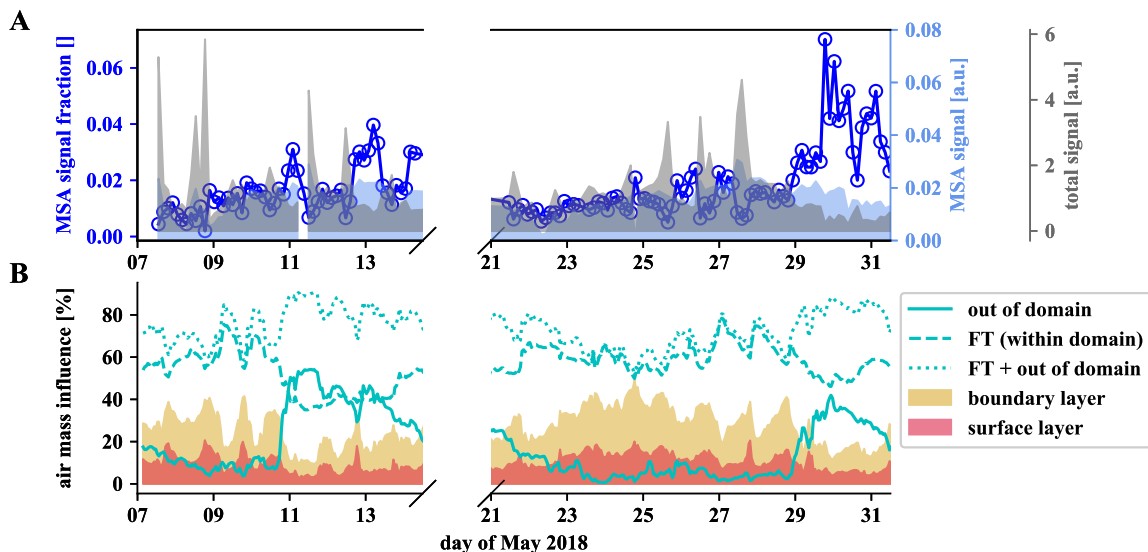

**Figure 7.** MSA from the particle phase, evaporated and detected by the I⁻-FIGAERO-CIMS (a) and FLEXPART analysis of the relative influence of the different air masses (b). A strong increase of the methanesulfonate fraction within the particles is observed when the influence of free tropospheric long-range transport increases.



### 3.5   The impact of local sources and long-range transport on the time series of DMS oxidation products

To understand which sources impact the DMS oxidation products' time series measured at CHC, we show the source regions ("footprints") of MSIA on a map together with the height-resolved domain lateral boundaries outside the white ring (see section 2.4 for an explanation), determined from FLEXPART Lagrangian dispersion calculations, as an example.

It is important to note, that this analysis is based on the full MSIA time series - not distinguishing between the free troposphere and the boundary layer. It shows source regions on the surface (inside the white ring) from which air masses likely traveled
close to the surface and height-resolved source regions at the lateral boundaries (outside the white ring). We chose to show the footprint analysis of MSIA for multiple reasons: first, we wanted to show a compound with a short atmospheric lifetime on the order of a few hours, which gives more weight to local sources compared to distant ones, thus giving an upper limit for the influence of local sources. Second, the time series had to be above the limit of detection for most of the time to have as many valid data points as possible, which unfortunately excluded DMS. Finally, the PTR3 sensitivity of MSIA is at the kinetic limit,
thus its time series is independent of humidity changes.

One of the resolved local source regions affecting the MSIA time series is the southern part of Titicaca lake, which also shows enhanced chlorophyll concentrations in May 2018, as shown in Fig. 8. More details on lake Titicaca are given in Sec. 2.1. The DMS flux from lake Titicaca has not been determined so far, but DMS fluxes from other lakes are roughly 10-times lower than the average marine flux of DMS (Steinke et al., 2018; Ginzburg et al., 1998; Sharma et al., 1999).

Compared to Titicaca Lake, the coast of Peru is a more important MSIA source according to the FLEXPART analysis - also

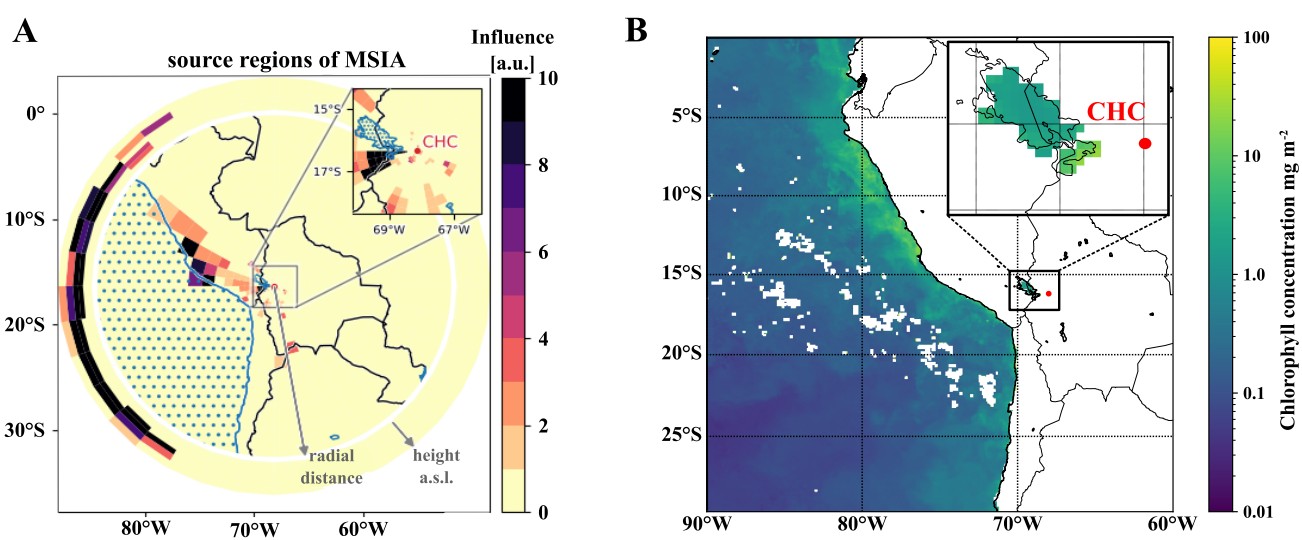

**Figure 8.** FLEXPART source regions determined using the MSIA time series (a) showing a strong impact of transport through the FLEX-PART domain lateral boundaries. Additional source regions seem to be the south of Lake Titicaca and the coast of Peru, which show enhanced Chlorophyll concentrations (b) averaged from satellite data (AQUA/ Modis) for May 2018.




showing enhanced Chlorophyll concentrations in satellite images (see Fig. 8). Convection is very unlikely to occur at this cold and typically overcast coast of Peru in May, but the air masses might still move upslope thermally driven, thereby staying close to the mountain slope surface.

However, the FLEXPART backward dispersion analysis locates the major fraction of the MSIA source regions in the domain
at the lateral boundaries in heights between 6000-8000 m a.s.l., suggesting that the air masses that carry DMS and DMS oxidation products are advected into the domain from the west (south-west to the north-west) and thus originate from the (remote) Pacific. For the other DMS oxidation products, the signal fraction explained by long-range advection is larger than for MSIA, as summarized in Table 3.

The differences in the allocation of source regions by FLEXPART for the various oxidation products likely originate from the

**Table 3.** Relative influence of short-range transport of local emissions (SR, $< 100$ km), long-range transport within the FLEXPART domain (LR, $> 100$ km) and from sources outside of the domain (ooD, influx through lateral boundaries, $> 1600$ km, free tropospheric transport)

|  | Relative influence | | |
| --- | --- | --- | --- |
| Molecule | SR | LR | ooD |
| DMSO | 0.2 | 0.13 | 0.67 |
| MSIA | 0.28 | 0.24 | 0.48 |
| MTF | 0.21 | 0.22 | 0.57 |
| $CH_3SOH$ | 0.09 | 0.45 | 0.46 |
| $DMSO_2$ | 0.16 | 0.24 | 0.60 |

different chemical transformation processes, which the Lagrangian dispersion model neglects. Despite the associated uncertainty, however, one result of the FLEXPART analysis is clear: Local sources contribute around 20% to the full time series of DMS oxidation products, while most of the signal is attributed to long-range transport with the largest fraction from outside of the domain. We can imply for our case - having even only a small impact of Lake Titicaca on the total time series - that the concentration of DMS oxidation products measured in the FT air masses are a result of marine DMS emissions and long range
transport and oxidation along the trajectory.

However, as described by Achá et al. (2018), the algal blooms in Lake Titicaca are likely to become more frequent in the future. Increasing DMS fluxes from Lake Titicaca would probably enhance especially the concentrations of DMS and its intermediates at CHC when the station is within the boundary layer.



### 3.6 Relating concentrations of DMS and its oxidation products with each other and to long-range transport

The observed DMS concentrations in comparison with the concentrations of its reaction products relate to a long reaction time during the transport from the source regions to Chacaltaya: The DMS signal is very variable, ranging from below detection limit ($3 \cdot 10^5$ cm$^{-3}$) to maximally $1 \cdot 10^8$ cm$^{-3}$. We observed its highest concentration during the FT event on the morning of the 11$^{th}$ of May 2018. During this time, the air mass was coming from the west with high wind speeds. In fact, the air masses only require about 30 hours to travel from the 1600 km distant lateral boundary of the simulated domain to CHC. This

period is similar to the DMS lifetime of ca. 1.5 days in the atmosphere with respect to its main oxidant OH under halogen- and NO$_x$-free conditions (Chen et al., 2018). Typical DMS concentrations in 6000-8000 m a.s.l. over the ocean are on the order of $5 \cdot 10^7 - 1 \cdot 10^8$ cm$^{-3}$ (Thornton et al., 1992), but can reach up to $4 \cdot 10^8$ cm$^{-3}$ close to convectional updrafts (Thornton and Bandy, 1993; Thornton et al., 1992). Consequently, the maximum value measured here is plausible, if transport from a convective cell to CHC was on the order of the DMS lifetime. Interestingly, an extremely rare subtropical cyclone has been

observed in the southeastern Pacific Ocean, only hundreds of kilometers away from the Chilean coast shortly before (May 9$^{th}$) (NESDIS, 2018), that most likely impacted the DMS concentrations at CHC. During all other FT events, the DMS concentration was $\leq 2 \cdot 10^7$ cm$^{-3}$, so that the major part of the DMS was already oxidized due to long transport times compared to the DMS lifetime.

All observed oxidation products except DMSO$_2$ show gas-phase concentrations typically between $10^5 - 5 \cdot 10^6$ molecules

cm$^{-3}$. DMSO$_2$ is the only compound with gas-phase concentrations regularly in the $10^7 - 10^8$ molecules cm$^{-3}$ range. At air temperatures of approximately -10°C at 6000 m a.s.l. (assuming a dry adiabatic temperature decrease with altitude above the altiplano), DMSO$_2$ is volatile to semivolatile (see Appendix A). In the free tropospheric air masses, DMSO$_2$ will be far more abundant in the gas phase than in the particle phase. The measured mean gas-phase DMSO$_2$ concentration of approximately $2 \cdot 10^7$ cm$^{-3}$ in the free troposphere equals a mass concentration of 3.1 ng m$^{-3}$. This is comparable to the total MSA mass

concentration when taking the particle phase data from the filter measurements (see Fig. 6) into account (section 3.4). Due to the close molar masses of DMSO$_2$ and MSA, this implies similar amounts of sulfur in these two chemical states.

Such high DMSO$_2$ yields are in contrast to chemical models, where DMSO$_2$ yields are smaller or DMSO$_2$ formation is even neglected completely (Hoffmann et al., 2020), due to previously reported low concentrations of DMSO$_2$ in the marine boundary layer (Davis et al., 1998; Berresheim et al., 1998). High gas-phase DMSO$_2$ concentrations were observed in lab

experiments and were attributed to the reaction of DMSO with O$_3$ at interfaces and in the aqueous phase (Berndt and Richters, 2012; Enami et al., 2016; Ye et al., 2021; Shen et al., 2022). The persistently high DMSO$_2$ concentrations compared to other DMS oxidation products observed at Chacaltaya might therefore also indicate oxidation at air-liquid interfaces of aerosols or cloud droplets.

All other oxidation products were significantly less abundant, which likely has to do with their shorter lifetimes: The DMSO/DMSO$_2$

ratio is relatively low, approximately 0.1. This is because the DMSO lifetime is short, typically hours, although DMSO is the precursor of DMSO$_2$ under low-NOx conditions. The ratio between MSIA and DMSO$_2$ is 0.03. Both compounds are products of DMSO oxidation, but MSIA has a similarly short lifetime like DMSO, while DMSO$_2$ is photochemically inert with




a lifetime of weeks and therefore builds up with reaction time along the trajectory. MTF (might be underestimated to some extent due to fragmentation), formed via H-abstraction and reaction with $HO_2$ and OH, thus requires two-step oxidation and

is an intermediate product. It has an at least 5 times higher concentration than MSIA. This is reasonable because MTF also has a longer lifetime than MSIA: Its lifetime is on the order of 6-12 hours due to its lower photolysis and reaction rates (by OH radicals), while it is around 3 hours for MSIA. MSA is detected at the kinetic limit in the nitrate-CIMS. DMSO, $DMSO_2$ and MSIA are detected at the kinetic limit in the PTR3 (see the collision induced dissociation ramps of DMSO, $DMSO_2$ and MSIA in the $H_3O^+$-CIMS in Shen et al. (2022)), so that this discussion is not impacted by uncertain calibration factors.

Furthermore, we measured signals close to the detection limits of our instruments at the exact masses of $CH_3S(O)_2OOH$ (in nitrate-CIMS) and $CH_3SOH$ (in the PTR3), that both have unknown reaction rate constants and unfortunately also uncertain sensitivities. According to Berndt et al. (2020), the latter reacts quickly with ozone.

## 4 Conclusions

From the SALTENA campaign at the GAW station Chacaltaya (CHC) in the Bolivian Andes, we extracted periods correspond-

ing to the prevalent influence of westerly free tropospheric air from the Pacific region to study marine-related chemical species transported over long distances.

Substantial signals of dimethyl sulfide (DMS) oxidation products were observed from all deployed state-of-the-art mass spectrometric techniques measuring both gas (APi-TOF, nitrate-CIMS, PTR3) and particle phase (ACSM and $I^-$-FIGAERO-CIMS). We observed DMS itself and its oxidation products $H_2SO_4$, MSA, DMSO, $DMSO_2$, MSIA, MTF, $CH_3S(O)_2OOH$

and $CH_3SOH$, of which several are highly soluble, showing the presence of DMS oxidation in the free troposphere. The array of time-of-flight mass spectrometers using different ionization methods and the low limit of detection allowed for the quantification of DMS, dimethyl sulfoxide (DMSO), dimethyl sulfone ($DMSO_2$), methane sulfinic acid (MSIA), methane sulfonic acid (MSA), and sulfuric acid in parallel.

We have used a Lagrangian backward dispersion model (FLEXPART) to determine the source region of the DMS oxidation

products within a domain of a 1600 km radius, which identified influx through the lateral boundaries in the free troposphere above the Pacific Ocean as the major source, suggesting an impact of convective updrafts from outside the FLEXPART domain. The model domain mainly covers the cold ocean waters affected by the upwelling Humboldt current that favor thermal inversion. Therefore these areas do not appear as sources of DMS reaching the Chacaltaya station.

The free tropospheric ion composition showed a strong reduction of amines compared to that in boundary layer conditions,

allowing DMSO to get protonated. $DMSO_2$ was found to have the highest concentrations of all DMS oxidation products in the gas phase. Gas-phase MSA concentrations reached up to $2 \cdot 10^6 \, \text{cm}^{-3}$ whenever the air masses exhibited especially low particle loads and water vapor mixing ratios, enhanced by a factor of 10 compared to the median MSA concentration. Nonetheless, even during periods with high condensation sink, we still found MSA in the particulate phase, around $3 \, \text{ng} \, \text{m}^{-3}$, which is comparable to the amount of $DMSO_2$ in terms of DMS-derived sulfur reaching CHC. Gas-phase MSA is therefore a good

tracer for times when long-range transport from the ocean and minimal influence from local surface emissions co-occur, while




the long-term data set of particulate MSA indicates that the station is regularly influenced by air masses with marine origin. The effect of DMS oxidation on particle composition is nonetheless small as it is competing with other sources of $H_2SO_4$ like volcanic $SO_2$ in the region. This made it impossible to relate the MSA to $H_2SO_4$ ratio to DMS oxidation conditions.

While previous measurements already suggested DMS oxidation in the free troposphere, such a full spectrum of compounds
from DMS oxidation as presented here has not been observed previously at such altitude and distance from sources, which provides insights into the fate of DMS in the free troposphere. Clearly, due to dilution and (wet) deposition losses, the concentrations of DMS oxidation products detected this far away from their sources are small. Our data suggest that their concentrations are too small to have an impact on new particle formation or growth directly at CHC, but - as previous results from flight campaigns have shown - particle formation and growth likely occur along the way while concentrations of MSA and DMS-
derived sulfuric acid are still less diluted and the air masses are not impacted by volcanic emissions. To properly quantify the impact of DMS oxidation in the free troposphere, laboratory studies on particle formation rates with varying MSA and sulfuric acid concentrations and flight campaigns closer to marine updrafts with modern low-detection-limit proton transfer and nitrate cluster reaction mass spectrometers are needed in combination with physical and chemical aerosol characterization.



*Code and data availability.* The time series of the different presented variables and the python codes for plotting the figures can be found
under doi.org/10.5281/zenodo.6866115 (Scholz et al., 2022)

## Appendix A: Volatility of $DMSO_2$

Dimethyl sulfone, $DMSO_2$, is a solid under typical atmospheric conditions in its bulk form. Only from 382 K onwards its vapor
pressure data are available, so that we need to estimate it. Using the data for $DMSO_2$ as obtained from National Institute of
standard and technology web thermo tables (NIST WTT) at the lowest available temperature (https://wtt-pro.nist.gov/wtt-pro/,
$T^o = 382$ K, $p^o = 0.7384$ kPa and $\Delta H_{vap} = 66 \, \text{kJ mol}^{-1}$) and

$$C = M \cdot \frac{n}{V} = M \cdot \frac{p}{RT} \tag{A1}$$

we find that $C^o = M \cdot \frac{p^o}{RT^o} = 94 \cdot \frac{738.4}{8.3 \cdot 382} = 21.9 \, \text{g m}^{-3} = 2.2 \cdot 10^7 \, \mu\text{g m}^{-3}$. The temperature dependence of the equilibrium
gas phase concentration is approximately

$$\log_{10}(C(T)) = \log_{10}(C^o(T^o)) + \frac{\Delta H_{vap}}{R \cdot \ln(10)} \tag{A2}$$

according to the Clausius-Clapeyron equation. With the $DMSO_2$-specific constants from above, this gives approximately
$C(T = 263 \, \text{K}) = 1776 \, \mu\text{g m}^{-3}$, which classifies $DMSO_2$ as a volatile to semi-volatile compound (Donahue et al., 2011, 2012).
It will therefore not contribute to particle formation, but might partition into the liquid aerosol phase due to its high water
solubility. This is independent of the droplet acidity according to De Bruyn et al. (1994).





## Appendix B: Location of the measurements in South America

Figure B1 gives a broad overview over the location of measurement within South America. Located close to the ridge of the Andes east of the Bolivian Altiplano at an Altitude of 5200m a.s.l., the station can be reached by free tropospheric air masses from the Pacific Ocean during the dry season, Amazon-influenced airmasses during the wet-season, as described in Bianchi et al. (2021).

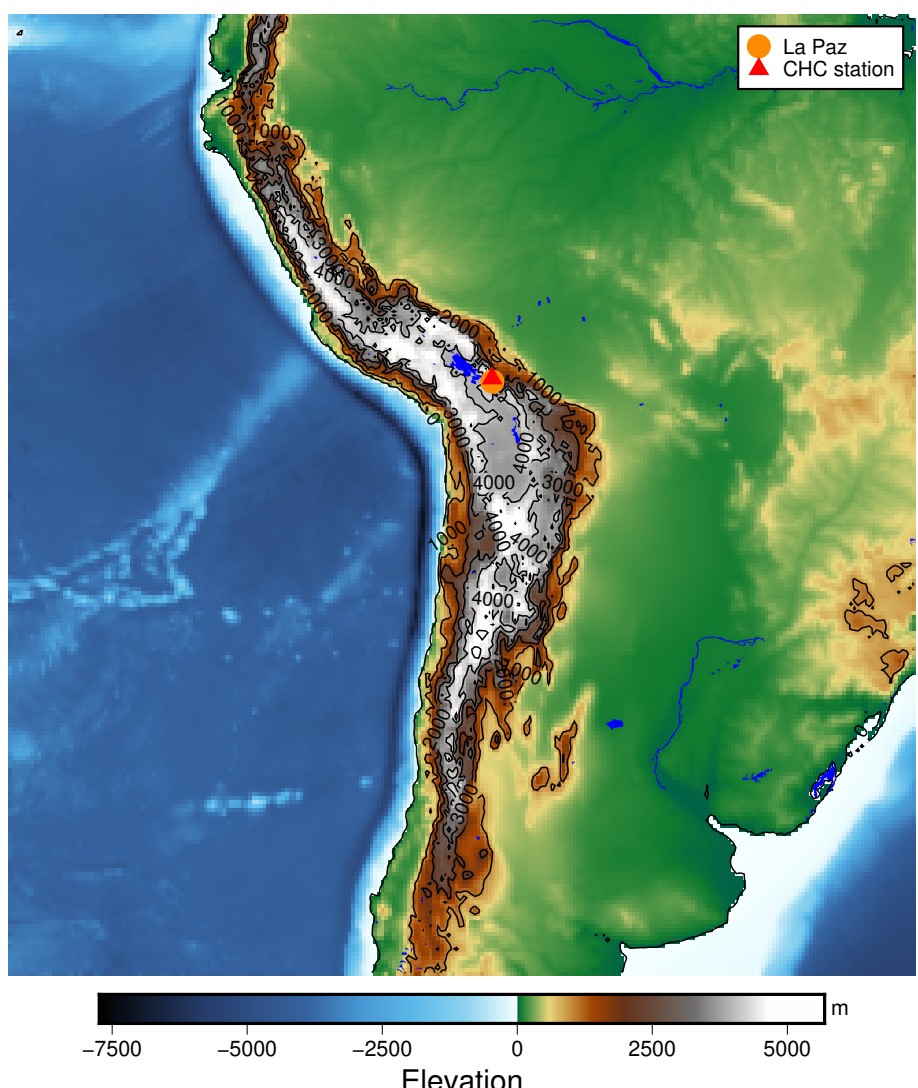

**Figure B1.** The location of Chacaltaya Global Atmosphere Watch (GAW) Station in the Bolivian Andes. The elevation data shown are taken from Tozer et al. (2019).




## Appendix C: Additional information on data quality

**C1 DMS calibration**

The settings of the PTR3, described in section 2.2.2 lead to a slightly humidity-dependent primary ion distribution, as shown in fig. C1. No larger clusters than $(H_2O)_2H_3O^+$ are stable at the set E/N value of $106 \pm 25$ Td. Therefore, the sensitivity towards molecules that can be ionized efficiently by all of $H_3O^+$, $(H_2O)H_3O^+$, and $(H_2O)_2H_3O^+$, does not exhibit a humidity-dependence. Molecules like DMS, have a humidity-dependence and need to be calibrated, because their gas-phase basicity

and proton affinity are not sufficient to get ionized by all of the clusters. The humidity-dependent sensitivity of DMS is shown together with that of Acetonitrile in fig. C2.

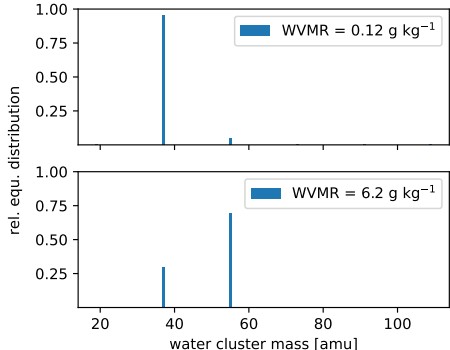

**Figure C1.** Calculated equilibrium (equ.) primary ion distribution in the PTR3 at the minimum and maximum water vapor mixing ratio (WVMR) inside the PTR3 during the sampling period.

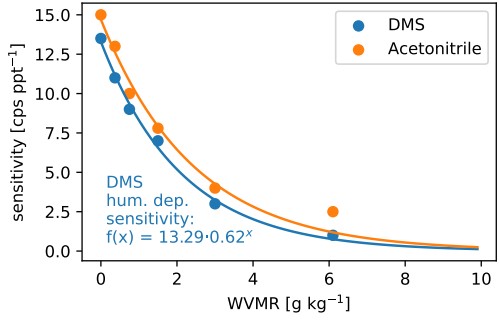

**Figure C2.** Water vapor mixing ratio (WVMR) dependent calibration of DMS after the campaign together with the reference acetonitrile that we calibrated regularly during the campaign. The DMS calibration data points are fitted with an exponential decay function (shown) that we used to calibrate the DMS trace.





## C2  measured gas-phase concentration distributions compared to the detection limits of the DMS-related molecules in PTR3 and Nitrate-CIMS

As concentrations in the free troposphere are generally small, a careful test of the limit of detection above which concentrations

are trustworthy, needs to be performed. In a mass spectrum, each signal peak can have another limit of detection, as it depends also on the background signal on the exact mass of the compound of interest and its stability. This background can be caused by other molecules (or their isotopes) with a similar exact mass, that might show a different daily pattern. The limit of detection was thus estimated for each of the compounds of interest, by taking into account any crosstalk of neighbouring peaks in the mass spectrum.

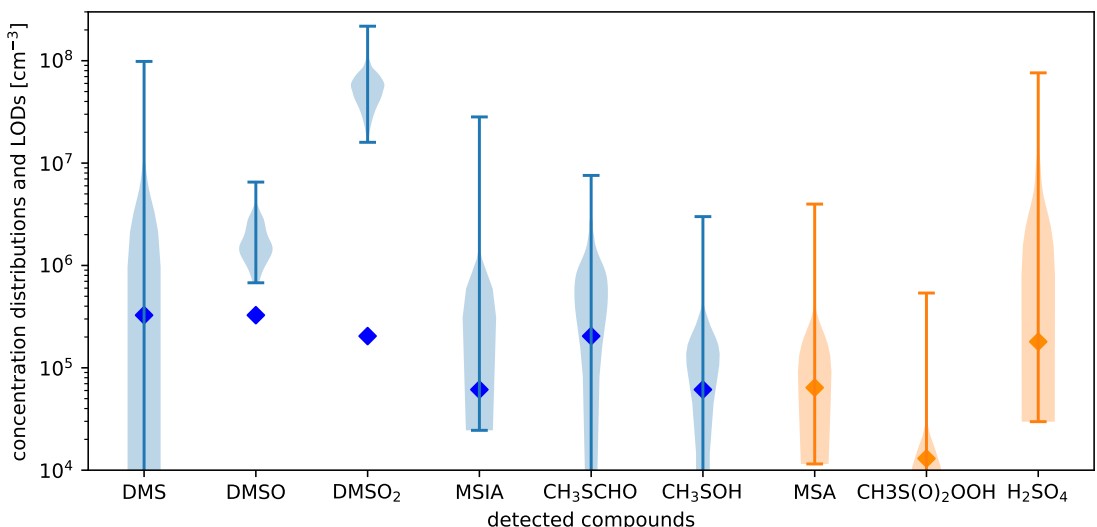

**Figure C3.** Observed gas-phase organosulfur compounds from PTR3 (blue) and nitrate-CIMS (orange) with their (lower limit) concentration distributions (violins) and limit of detections (LODs, diamonds)



# Appendix D: Additional information on gas- and particle phase, May 2018

In this appendix, we have collected some figures that may help the interested reader to better understand the development process of our results and conclusions presented in the main section.

Fig. D1 shows the time series of the gas-phase concentrations of the different observed molecules belonging to the DMS scheme, which are also available in the Supplementary Material as a data set (see *Code and data availability*).

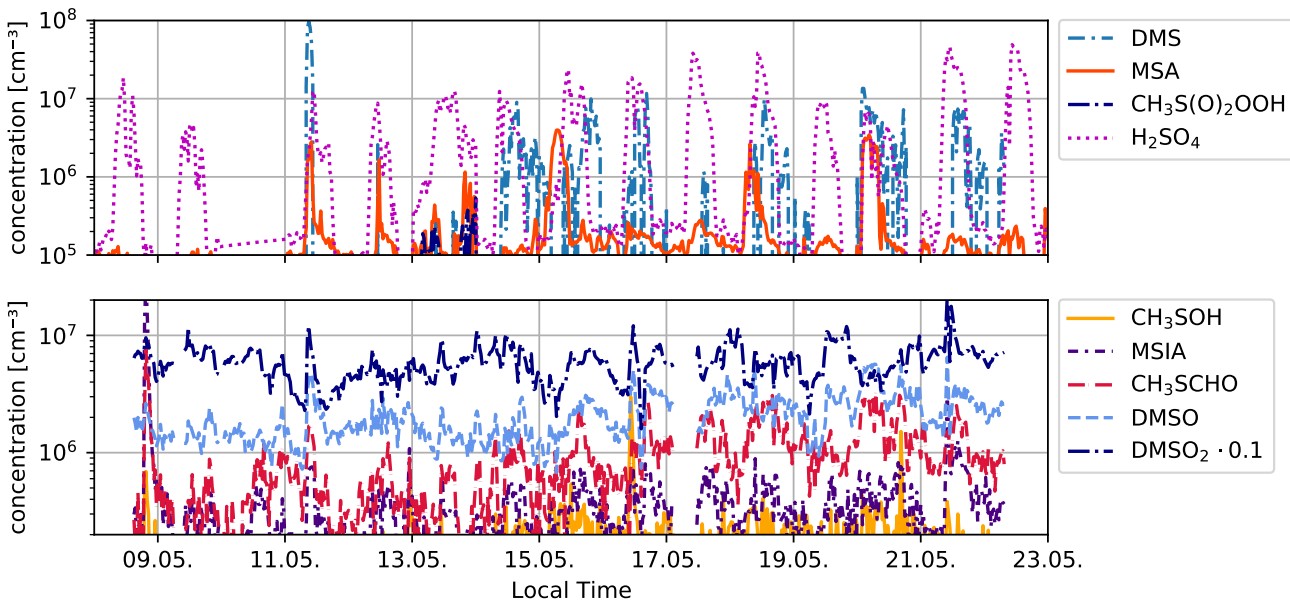

**Figure D1.** Time series of all detected compounds (measured by PTR3 and nitrate-CIMS) that belong to the DMS oxidation scheme during two weeks of May 2018.

## D1  Comparison with boundary layer compounds

Fig. D2 allows to relate the FT and BL dependence of DMS and its oxidation products in fig. 5 in the main to the usual behavior of boundary layer-affected measurands. Here we would also like to reiterate that nearly all of the gas-phase molecules observed in the mass spectra of nitrate-CIMS and PTR3 clearly follow the pattern of boundary-layer-affected substances and that the signals within the boundary layer are typically orders of magnitude higher during the day than in the free troposphere, while this is not true for the DMS and its oxidation products. This is also evident in Fig. D3, which shows spectra of the nitrate-CIMS in the boundary layer (downward) and in the free troposphere (upward). The latter is clearly dominated by MSA.





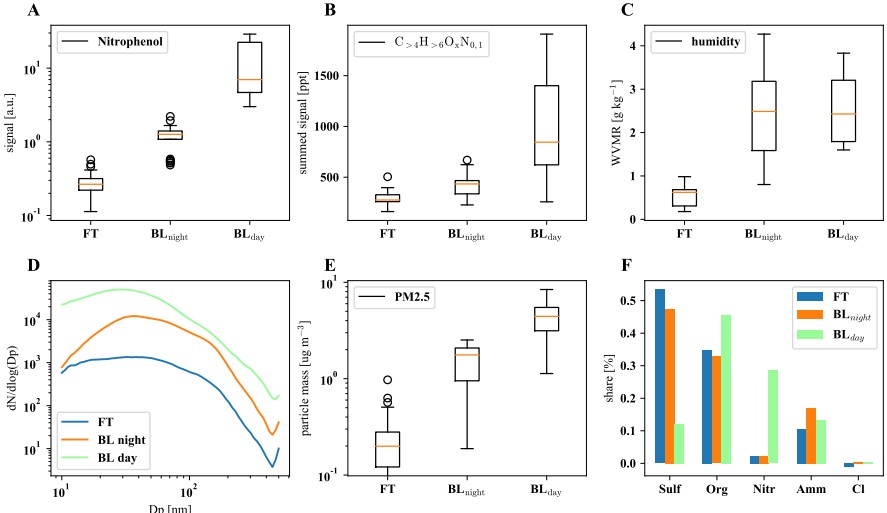

**Figure D2.** The first row shows Nitrophenol (A), total organics $C_{>4}H_{>6}O_xN_{0,1}$ (B) and water vapor mixing ratio (C), the second row summarizes particle size distribution (D), total $PM_{2.5}$ mass (E), and composition (F).

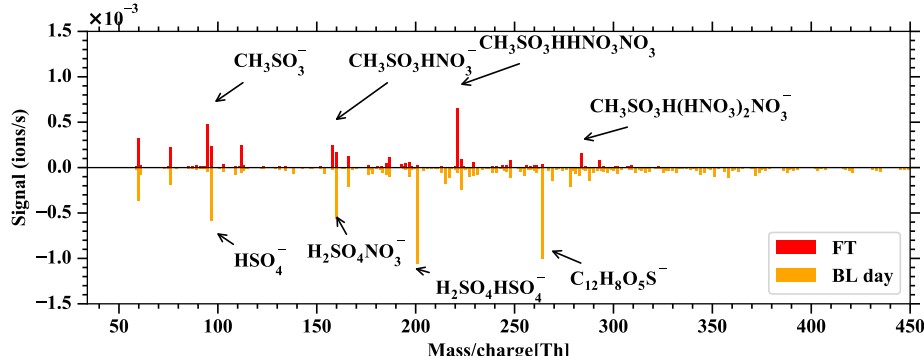

**Figure D3.** Spectra of the nitrate-CIMS, showing the difference FT - $BL_{day}$ on the positive axis and $BL_{day}$ - FT on the negative axis. Only very few compounds are larger in FT than $BL_{day}$: The positive spectrum is dominated by clusters of the primary ions with MSA.

## D2 Nighttime sulfuric acid and the condensation sink

In fig. 5 in section 3.3 we have shown that the concentration of sulfuric acid in the free troposphere is high compared to the night-time sulfuric acid in the boundary layer. While parts of the free tropospheric (FT) periods are after sunrise, we see in fig. D4, that the sulfuric acid concentration is also high during night-time FT periods, such as the night of the 20th of May. This is likely connected to the low condensation sink, that we show together with the sulfuric acid in the same figure.

To calculate the condensation sink, we followed the approach described in Leskinen et al. (2008) with mean free path, diffusion





coefficient and size resolved particle number concentration at 500 mbar pressure. Because the diffusivity is antiproportional and the number concentration proportional to the pressure, they cancel each other and we can use the diffusivity and the

605 number size distribution both given at 1013 mbar standard pressure. For the diffusivity we used D=$6.5 \cdot 10^{-6}$ m$^2$s$^{-1}$, the mean literature value for sulfuric acid, extracted from the compilation by Brus et al. (2016) for 278K. The mean free path has to be pressure-corrected however, as it does not cancel out. For 500 mbar we used a mean free path of 138 nm.

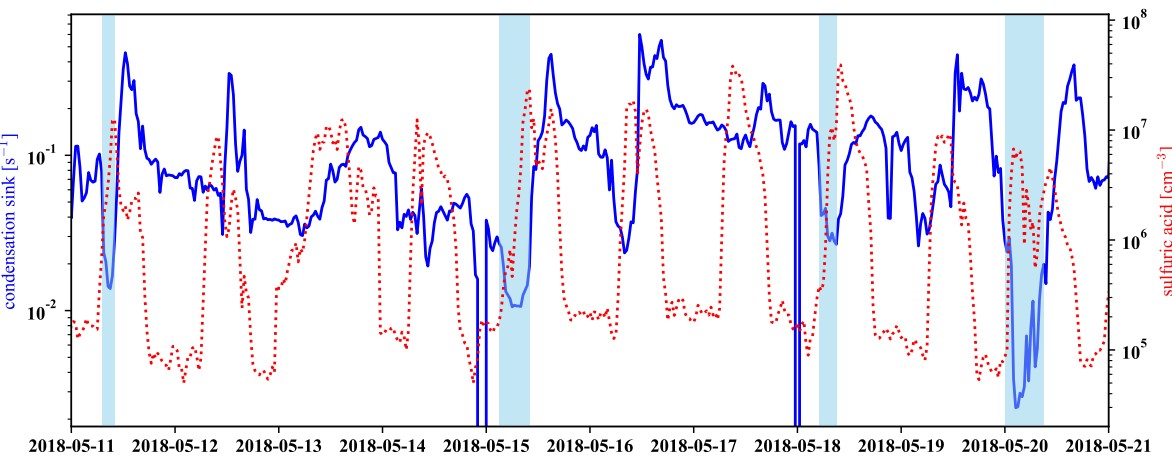

**Figure D4.** Time series of sulfuric acid in red and of the condensation sink (pressure-corrected) in blue, calculated as described below. Marked in light-blue are the identified FT periods.



## Appendix E:  Complementary information for the discussion on ion composition, February, 2018

Fig. E1 shows additional information for the discussion in section 3.2 where we presented the time evolution of positively
610  charged DMSO clusters under different air masses. However, other positively charged compounds such as organics are not
presented in the time series, and are instead displayed in mass defect plots in Fig. E1, which focus on the difference in ion com-
position between the boundary layer and free troposphere by extracting the periods corresponding to the prevalent influence.
The night-time ion spectrum in the boundary layer (figure E1 A) is dominated by amine groups (coloured circles) including

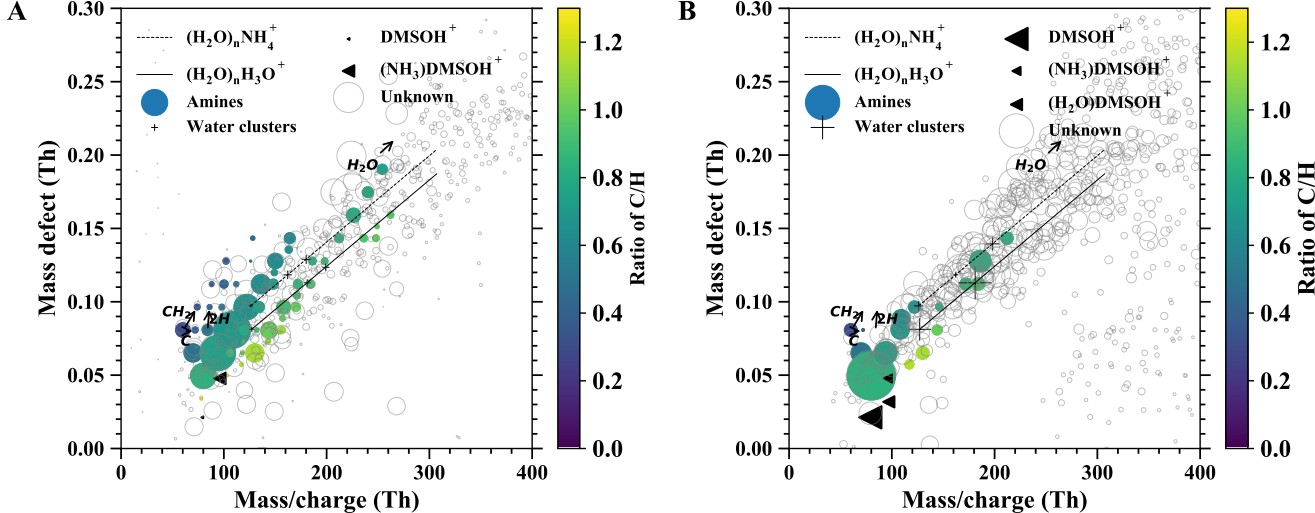

**Figure E1.** The mass defect for airborne ions in the continental boundary layer (A) and the free troposphere (B). The data are averaged over
1 hour.

alkyl propyl amines ($(CH_3)_3(CH_2)_n NH^+$ ), alkyl pyrroline ($C_4H_7(CH_2)_n NH^+$), alkyl pyridines ($C_5H_5(CH_2)_n NH^+$), alkyl
pyrrolidine ($C_4H_{10}(CH_2)_n NH^+$), alkyl didehydropyridine ($C_5H_4(CH_2)_n NH^+$), alkyl indole ($C_8H_7(CH_2)_n NH^+$), alkyl ben-
zazocine ($C_{11}H_9(CH_2)_n NH^+$), alkyl quinoline ($C_9H_7(CH_2)_n NH^+$), etc. Besides the amines, charged water clusters and
water ammonia clusters are the second major species. This is likely due to their abundant concentration in the boundary layer,
although water and ammonia have lower proton affinity than the amines. The charged water and water-ammonium clusters
usually contain 6 to 11 water molecules according to the ion spectrum in Figure E1. However, we are not able to observe all
the clusters in the ion spectrum because some of them interfere with the species that have similar mass-to-charge ratios. For
example, $(H_2O)_7 \cdot NH_4^+$ is not visible due to $C_{10}H_{10}N^+$, which has a stronger signal as shown in Figure E1 A.
The air mass origins shown in Fig. E2 help to confirm that our ionic data set is comparable with the free tropospheric periods
during the intensive measurement period in May.

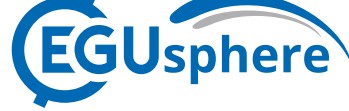



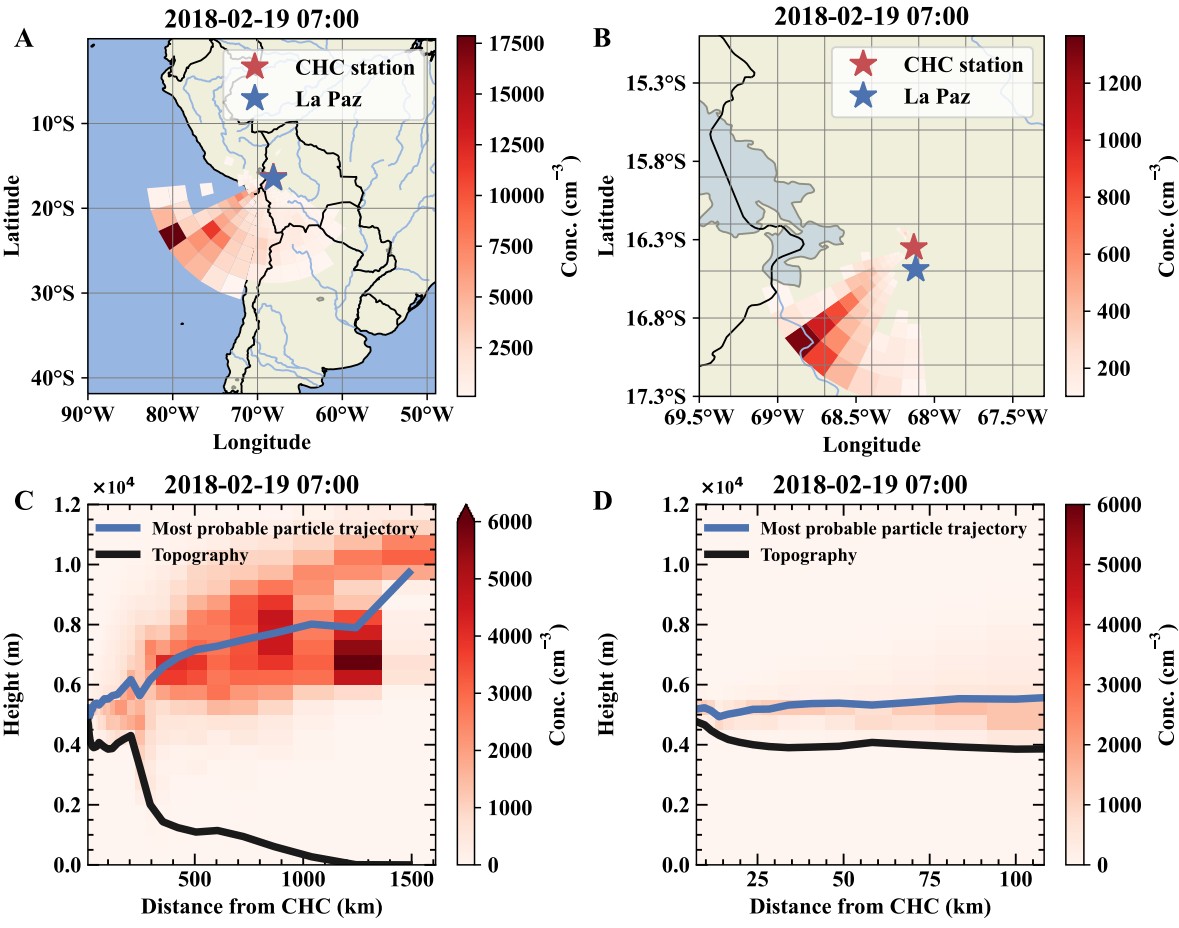

**Figure E2.** Air mass origin of a free tropospheric event on 19[th] February. Panels A and B show the top-down view of the most likely source regions for the air parcels from west-southwest, which is close to the conditions on 11[th] of May. Panels C and D show the same data but height resolved together with the topography as a vertical cut through the atmosphere.



*Author contributions.* W.S., Q.Z. and J.S., C.W., S.C. and P.A. analysed the data of the mass spectrometers PTR3, nitrate-CIMS and Api-
TOF, I⁻-FIGAERO-CIMS, and Q-ACSM, respectively; D.A. performed the FLEXPART lagrangian dispersion simulations and analysis;
J.L.J. analyzed the long term aerosol filter samples with support from P.G. and G.U.; W.S., D.A., C.W., S.C., I.M., Q.Z., W.H., L.H., J.L.J.,
E.P., M.L., F.V., C.M., F.B. collected the data and operated the instruments during the measurement campaign; P.L. and M.A. cofounded the
CHC GAW station and oversee the atmospheric research at the station since the beginning. P.G. and G.U. provide organizational and financial
support for the long-term research. P.G. also ensured custom clearance of all instruments involved in the SALTENA campaign. W.S. and J.S.
wrote the manuscript with contributions from V.S.; contributed to data interpretation and editing of the manuscript. All authors commented
on the manuscript.

*Competing interests.* There are no competing interests to declare.

*Acknowledgements.* We thank the Bolivian staff of the IIF-UMSA (Institute for Physics Research, UMSA) who work at CHC for their
valuable work under difficult conditions and the IRD (Institute for Research and Development) personnel for the logistic and financial
support during all the campaign including shipping and customs concerns. We also acknowledge the CSC — IT Center for Science, Finland,
for generous computational resources that enabled the WRF and FLEXPART-WRF simulations to be conducted. We thank the European
Union (EU) H2020 program via the fundings European Research Council (ERC; project CHAPAs no. 850614 and ATM-GTP no. 742206)
and the Marie Skłodowska Curie (CLOUD-MOTION no. 764991), the Finnish Centre of Excellence as well as the Academy of Finland
(project no. 311932, 315203 and 337549), the Knut and Alice Wallenberg Foundation (WAF project CLOUDFORM no. 2017.0165). P.
Artaxo acknowledges funds from FAPESP – Fundação de Amparo à Pesquisa do Estado de São Paulo, grant 2017/17047-0. W. Scholz
thanks the University of Innsbruck for their support in form of a doctoral scholarship (2021/01).
The long-term observations used in SALTENA are performed within the framework of GAW and ACTRIS, receiving support from UMSA,
and from the international stakeholders. In France, support from CNRS through ACTRIS-FR/SNO CLAP, and IR DATA TERRA, Institut de
Recherche et Développement (IRD) and Observatoire des Sciences de l'Univers de Grenoble (OSUG) through ANR LABEX in particular
is greatly acknowledged. All long term measurements of the filters chemistry were performed on the Air O Sol analytical plateform at IGE
(Grenoble).
We acknowledge the use of imagery from the AQUA MODIS satellite, provided by services from NASA's Global Imagery Browse Services
(GIBS), part of NASA's Earth Observing System Data and Information System (EOSDIS).



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
