# Peer review of "Measurement Report: Long-range transport and fate of DMS-oxidation products in the free troposphere derived from observations at the high-altitude research station Chacaltaya (5240 m a.s.l.) in the Bolivian Andes"

_EGUsphere, 2022_

## Author Response (AR1)

Dear Editor,
Dear Reviewers,

Thank you for your helpful guidance and responses to our manuscript.
This response on the discussion of Preprint egusphere-2022-887 is structured and marked as follows:

**Structure:**

**Reviewer Comments marked in bold black**
Followed by our response marked in blue
*and changes in our manuscript in italic, showing if something is*  *added*
* * *
**Response to Reviewer 1:**

**The paper presents a very rare high altitude data set of state-of-the-art measurements of DMS and its oxidation products from CHC in the Bolivian Andes. A modified FLEXPART scheme was used to identify source regions of the chemical species measured and the degree of influence of the boundary on the measured gas and aerosol phase composition. Strong evidence is presented for long range transport of DMS and oxidation products in the free troposphere across the Pacific to CHC.**

Dear Colleague, thank you very much for your detailed assessment of our manuscript and for your suggestions on readability. We will now answer your comments one by one.

**Line 110: Why was the figure of the measurement site put into Appendix B rather than the main text? It would be useful to have it in the same section as the station description for those not familiar with the geography surrounding CHC. Also, please label the location of Lake Titicaca and the other lagoons mentioned in line 121.**

We moved Fig. 8B (now Fig. 1) into the section on the location of CHC, further Lakes and Lagoons are shown as suggested:

[Figure]

*Figure 1. The location of the Chacaltaya GAW station (CHC) with a focus on potential DMS emission hotspots in the region. The color code shows the average Chlorophyll concentration in the Pacific Ocean and Lake Titicaca as shown in the enlarged insert for May 2018 as obtained from satellite data (AQUA/Modis). All lagoons are marked with darkblue outlines. The horizontal extent of the model domain for Lagrangian backward calculations (see sec. 2.4 for details) is depicted as a dashed, pink line.*

**Table 1: The Q-ACSM species should be labelled as non-refractory in the table.**

We corrected this according to your suggestion. Table 1 entry for the Q-ACSM now reads:

*Instrument      detected species*

*Q-ACSM        summed non-refractory organics, nitrate, sulfate, ammonium, chloride*

**Line 142: It is stated that the nitrate-CIMS made measurements from April 19th... to 25th but based on what is shown in Figure 1, it looks like those data were not analyzed. An explanation of the reason why not all data was analyzed (for all instruments) would be useful.**

The nitrate-CIMS was operational for an extended period, as shown in Fig. 2 (former Fig. 1). However, due to technical difficulties and many power cuts during the wet season, the instrument was not performing always, so large fractions of the MSA data are rather uncertain. The 19th-25th of April would be of good quality. We however decided not to go into further detail with these data, because – as mentioned a few times throughout the manuscript - we focused our analysis on periods of typical dry season conditions (clear sky, westerly winds) that were not met in that time period. Maybe the word "analyzed" is not well-suited, as we have looked into all data, while this paper only focuses on the marked periods. To clarify this, we will add a note in the caption of Figure 2 and change the word "analyzed" to "herein presented" data.
The caption of Fig. 2 is adjusted (see next point) and also lines 136 ff:

*Table 1 summarizes which of the analyzed species are detected by which instrument. An overview of the operation times of the instruments and the  herein presented periods is given in Fig. 2.*

**Line 200 says the FIGAERO-CIMS was operational April 10th... to June 2nd but Figure 1 indicates it became operational in early May. Vertical lines corresponding to dates on the x-axis would help.**

Yes, the inconsistency probably occurred due to a misunderstanding within the group, that was resolved shortly before submission, and we forgot to update the figure. The dates given in the text in version 1 were correct. Now Figure 2 (former Fig. 1) is adjusted accordingly**:**

[Figure]

*Figure 2: Overview over the operation times of the individual mass spectrometers and the herein presented time periods. The herein presented times are chosen by filtering for typical dry-season conditions (westerly winds, clear sky) and removing all periods with uncertain data quality, e.g. due to technical difficulties and power cuts.*

The reason for not focusing on the FIGAERO data from April is - as mentioned above - that our aim was to focus on typical dry-season conditions for this manuscript.

**Section 3.1: The different panels within Figure 2 should be described in the order they are mentioned in the text. Currently, the order is 2A, 2E, 2C, etc.**

The order of lines 274 ff are now adjusted to fit the order of subpanels in Figure 3 (former Fig. 2):

*Fig. 3 B shows the amount of air coming from the free troposphere (>1500 m a.g.l.) and influenced by surface emissions (0-500 m a.g.l.) according to the results of the FLEXPART lagrangian dispersion analysis. Furthermore, we show the fractions of free tropospheric air from "out of" the FLEXPART domain and "within the domain" (defined in sec. 2.4).*

*We also present the water vapor mixing ratio (WVMR) and equivalent black carbon mass (eBC) (Fig. 3 C), as well as short wave radiation (SWR), and temperature (Fig. 3 D), Q-ACSM total particle mass and its composition (Fig. 3 E), followed by the particle number size distribution (Fig. 3 F).*

*Fig. 3 B reveals that free tropospheric air masses reach the CHC GAW station all the time during the period shown. However, neglecting the varying boundary layer height in these traces (see Aliaga et al., 2021, section 4 for a justification of the differentiation using constant pseudo layer heights instead of a time-resolved boundary layer height)*  *leads to a tendency to underestimate the surface influence during daytime and the free tropospheric air during nighttime (when the inversion is below 1500 m a.g.l.).*

*Especially during the daytime, free tropospheric air mixes with…*

**Lines 274 – 275: It doesn't seem necessary to say that they "appear anticorrelated". Can't the regression be performed to see what the degree of anticorrelation is?**

**"Anticorrelation" might be the wrong word,** as we have no close-to-linear relationship between these data, but the scatter plot inserted below and added into the paper as new Fig. D4 (former Fig. D4 became Fig. D5) shows that the processes leading to high MSA vs. high summed $C_8H_{10}O_x$, are **counteractive.**

Additions to the appendix:

*Figure D4: Correlation plot of MSA and $C_8H_{10}O_x$ showing that MSA concentrations are only high when corresponding $C_8H_{10}O_x$ values are low.*

Adjustment in the main part of the paper (lines 273 / 274):

*It is noticeable, that gas-phase MSA  is high whenever the $C_8H_{10}O_x$ compounds are low, as shown more clearly in Fig. D4.*

**Lines 297 – 302: The panels within Figure 3 are also mentioned out of order in the text.**

The order of lines 303 ff. is now adjusted to fit the order of subpanels in figure 4 (former Fig. 3):

*The FLEXPART dispersion simulation  can resolve this, as shown in Fig. 4 for the 11th of May,  as an example: During the whole day, the prevailing wind direction was west-southwest (Fig. 4 A and B for the 8 am case). Fig. 4 C and D clearly show, that the air reaching CHC ...*

**Line 300: It is difficult to understand what is being said here since it is not clear what Figures 3 B1 and B2 are. Is this statement based instead on Figure 3F ("during the afternoon of the same day, shown in Fig. 3 B1 and B2, a still small, but larger fraction of the air travelled uphill close to the surface")?**

Yes, that must've slipped our eye. Figs. 4 (former Fig. 3) B1 and B2 are actually figs. 4 E and F. We corrected the reference to these subpanels in the new version:

*In contrast, during the afternoon of the same day, shown in Fig. 4  E and F, a still ...*

**Two methods are used for designating the sampled air masses as primarily FT with little influence from the boundary layer or influenced by the local boundary layer - the FLEXPART analysis and the value of the identifiers listed in Table 2. Some discussion of why two methods were used and the unique information each provided would be useful.**

In order to analyze whether or not the station is above the boundary layer at a given time, the thresholds of the identifiers are temporally more precise than the FLEXPART analysis:

The FT intrusions sometimes happen on a timescale, which is close to the limit of the model time resolution (1 hour), while the various identifiers typically have higher temporal resolution.

In addition, the identifier approach uses direct on-site measurements that are generally more reliable than the FLEXPART analysis, which builds upon the WRF model that does not represent all the complex mountain meteorology in the vicinity perfectly, although it does a very good job:

The model, when evaluated against short-range BL influences does nonetheless perform very well on average (see Bianchi et al 2021, doi.org/10.1175/BAMS-D-20-0187.1) and extended periods that are longer than the model time resolution. Fig. 4 E and fig. 9 show this performance and the agreement between these methods gives us confidence in both our FT identification and the FLEXPART model. However, for short single events, there are generally higher chances of the model missing the complex local meteorological features that drive the FT/BL interactions. As discussed in lines 280 ff., the constant pseudo boundary layer induces a tendency to underestimate the surface influence during daytime and the free tropospheric air during nighttime.

To summarize, for a temporally accurate subdivision into "above" and "within" the boundary layer, especially on short time scales, the indicators might be more appropriate, especially when aiming for a very strict identification of FT periods.

On the other hand, identification of the air mass origin with FLEXPART is an excellent tool for determining the most likely source regions of a compound (footprint analysis, see sec. 2.4 and 3.5), and it shows the air masses origin during a certain period, which is not possible with the FT indicators measured directly at the station. It also identifies the height at which the long-range transport occurred, which affects e.g. the temperature and thus chemical conversion during transport.

Finally, the FT identifiers (e.g. absolute humidity) might vary depending on the season and are set a-posteriori to data analysis, in contrast to the a-priori FLEXPART results.

We added this evaluation in short words into the manuscript in lines 312 ff.:

* To avoid uncertainties caused by the FLEXPART time resolution and the potential day-time-dependent over- and underprediction of the boundary layer influence, we also determine when the station is in the free troposphere above the shallow boundary layer by applying the conditions from Table 2.  These FT identifiers are chosen after careful data analysis to be very strict, so that we certainly exclude all periods affected by the boundary layer. We assume that all periods during which the conditions are not fulfilled, are likely influenced by the local boundary layer to at least some extent and we refer to them as $BL_{day}$ between 7 am and 7pm local time an $BL_{night}$ otherwise. The FLEXPART analysis provides us with additional information on the horizontal and height-resolved transport of the airmasses. Furthermore, the FLEXPART analysis can be considered a-priori to data analysis, while the FT identifiers are set a-posteriori and might vary with seasons (such as the water vapor mixing ratio).*

**Lines 312 – 315: It would be helpful to include a figure showing the FLEXPART domain when it is discussed in Lines 251 – 254.**

As you suggested, we moved previous Fig. 8B (now Fig. 1) to section 2.1 and included the horizontal extent of the FLEXPART domain as a pink dashed line there, that way a first idea of the extent is given to the readers early on. We now reference Fig. 1 in line 250.

**Line 314: Should be "Intertropical" Convergence Zone.**

Thank you for noticing, we corrected this.

**Throughout: please use uniform date-time stamps in the text and figure captions.**

Thanks for this suggestion.

For a good readability of all figures, we decided to go with the following formatting rule:

For figures covering multiple months, we only show the months in their abbreviated form (e.g. Jan, Feb, Mar…), mentioning the year. Whenever the period shown is shorter, we use the day of the

month as tick labels and mention month and year in the label of x-axis. Whenever more precise timestamps are needed, we provide these as minor ticks, labelled with HH:MM. Although this is not a completely uniform format for all plots, it still enhances the similarity between the figures and thus the readability compared to before.

These changes affected figures 2, 3, 5, 7, 8, D1, and D5.

**Figure 4: Again, panels are mentioned out of order in the text.**

Thank you for noting this. We considered changing it, but in some cases, such as this one, we believe that it is more helpful to have the plot set up in such a way, that most important variables are shown in the upper-most panel.

In contrast to the figure, it makes sense to build up the argument in the text, starting with something that is already known to the reader: Here, the FT identifiers. Therefore, we would like to leave the order as it is.

**Figure 5: How many data points are the boxplots constructed from? Would it be more appropriate to use an average and standard deviation?**

Each boxplot is based upon 32 data points of pre-averaged 30 min. intervals. The night- and daytime boundary layer conditions were chosen to be as close as possible (time-wise) to the FT periods with the same prevailing wind direction (horizontal air mass origin) for the best comparison.

We noticed, that this was previously mentioned too early within the text (lines 306 ff), and we moved the information now to figure captions of both Fig. 6 (former Fig. 5) and Fig. D2.

We think it makes sense to show the mean value as you suggested (see below). In a few cases (like DMS), only a few very high values impact the mean. As visible in Fig. 6, for many groups the data are skewed, so just giving mean and standard deviation is not as useful.

We added the mean values as green triangles to Fig. 6 and adjusted the caption by adding the following lines (in Fig. D2 accordingly):

[Figure]

*The plots are based on 32 data points from averaged 30-minute intervals, with nighttime and daytime boundary layer conditions chosen to be as close in time as possible to FT periods with the*

*same prevailing wind direction (horizontal air mass origin) for optimal comparison. The orange line shows the median, the green triangle the mean value of the data.*

**Figure 7 is mentioned in the text before Figure 6.**

We changed the order of appearance of former figures 6 and 7 accordingly. Be aware, that they are now, in fact, figures 7 and 8.

**Figure 7: Gridlines would help guide the eye to see correlations between peaks and valleys of the plotted parameters.**

Thanks, we added the grid lines as suggested.

**Figure 6: It would be helpful to add the monthly wind direction to the figure.**

The main message from the plot (be aware that former figure 6 is now figure 8) should be, that a stronger west wind influence enhances the MSA concentration measured. We therefore decided to show the frequency of wind direction data falling into the range from SSW to NW as an additional subplot. That way we keep the figure simple, but still fully support the message that we're making in this section:

[Figure]

*Figure 7. Yearly pattern of methanesulfonate mass concentration in PM2.5 and PM10 particles, based on data between 2012 and 2017, taken at CHC (A) and westwind (SSW to NW) frequency as measured on the Chacaltaya mountain top (B). In the months from December to March the wind is often coming from the east due to the Bolivian high (Bianchi et al., 2021). In these months methanesulfonate mass concentrations in the particles are significantly lower than during the rest of the year.*

**Figure 8 could be included earlier in the manuscript to show the FLEXPART domain and the location of CHC and Lake Titicaca.**

Thank you for this idea. We moved previous subfigure 8B to the methods (now Fig. 1). Previous subfigure 8A does now stand alone as Fig. 9.

**# Response to Reviewer 2:**

**This manuscript summarizes a large dataset of gas and aerosol data collected at the GAW station in the Bolivian Andes. The authors deployed five state of the art mass spectrometers to measure DMS and its oxidation products. The data are further supported by the regular station measurements. The authors used chemical markers and FLEXPART to eliminate local contamination and to select periods of free troposphere flow. The relative mixing ratios of DMS and its oxidation products could be explained in these periods of free troposphere flow based on the compound lifetimes. It is a very interesting data set. The paper is well written and supported by five appendices. I recommend publication as is.**

Dear Colleague, thank you very much for this positive feedback! While we do change our paper somewhat according to the comments of Reviewer 1 to make it even more readable and easier to follow (see above), the general style of writing and the points we are making throughout the manuscript certainly remain the same.

**# Response to Author Comment 1:**

**The reference to Hoffmann et al., 2020 in lines 491-492:**

**"Such high DMSO$_2$ yields are in contrast to chemical models, where DMSO$_2$ yields are smaller or DMSO$_2$ formation is even neglected completely (Hoffmann et al., 2020),"**

**is not fully correct. In their model, Hoffmann et al. only neglect liquid phase DMSO2 formation and further oxidation of DMSO$_2$, while gas-phase DMSO$_2$ production via DMSO + NO$_3$ and DMSO + BrO are included.**

This sentence has now been corrected to:

*Such high DMSO$_2$ yields are in contrast to chemical models, where DMSO$_2$ is simulated to be only a minor oxidation product of the DMS addition pathway in the marine boundary layer* *(e.g. Hoffmann et al., 2016, Hoffmann et al, 2020), due to previously reported low concentrations of DMSO$_2$ in the marine boundary layer (Davis et al., 1998, Berresheim et al., 1998). However, the colder temperatures, and the low humidity together with the long reaction time and negligible losses likely favor its accumulation in the free troposphere.*